# Parameter Estimation to Improve Coastal Accuracy in a Global Tide Model

Xiaohui Wang[1], Martin Verlaan[1,2], Jelmer Veenstra[2], and Hai Xiang Lin[1]

[1]Delft Institute of Applied Mathematics, Delft University of Technology, Delft, The Netherlands
[2]Deltares, Delft, The Netherlands

**Correspondence:** Xiaohui Wang (X.Wang-13@tudelft.nl)

**Abstract.** Global tide and surge models play a major role in forecasting coastal flooding due to extreme events or climate change. The model performance is strongly affected by parameters such as bathymetry and bottom friction. In this study, we propose a method that estimates bathymetry globally and the bottom friction coefficient in the shallow waters for a Global Tide and Surge Model (GTSMv4.1). However, the estimation effect is limited by the scarcity of available tide gauges. We propose to complement sparse tide gauges with tide time-series generated using FES2014. The FES2014 dataset outperforms GTSM in most areas and is used as observations for the deep ocean and some coastal areas, such as Hudson Bay/Labrador, where tide gauges are scarce but energy dissipation is large. The experiment is performed with a computation and memory efficient iterative parameter estimation scheme applied to Global Tide and Surge Model (GTSMv4.1). Estimation results show that model performance is significantly improved for deep ocean and shallow waters, especially in the European Shelf directly using the CMEMS tide gauge data in the estimation. GTSM is also validated by comparing to tide gauges from UHSLC, CMEMS, and some Arctic stations in the year 2014.

## 1 Introduction

Accurate prediction of water levels in coastal areas is of significant importance. Coastal flooding, mainly caused by storm surges, is one of the main risks for the world's coastal (McGranahan et al., 2007; Kron, 2012). Global exposure to flooding has had an upward trend in recent years due to climate change, and sea-level rise (Hallegatte et al., 2013; Wahl et al., 2017; Oppenheimer et al., 2019). For example, the global sea level is currently rising at 3-4mm/year, and a 10-20 cm sea-level rise would more than double the frequency of coastal flooding before 2050 (Vitousek et al., 2017). This demands global tide and surge models that can provide sea-level estimates for the large-scale assessments of the flooding risk (Ward et al., 2015).

Global tide models are often divided into three groups, empirical tide models, purely hydrodynamic forward models, and hydrodynamic tide models with data assimilation (Stammer et al., 2014). To study the interaction of tides with other processes such as surge and sea level rise, it is very useful to model tide and surge together in one model. For example, the Global Tide and Surge Model (GTSM) is not only capable of simulating tides but also surges by adding meteorological wind and air pressure forcing (Verlaan et al., 2015). Another application is the provision of boundary conditions for regional models (Zijl et al., 2013). The accuracy of tide and surge models has improved significantly over the past decades, through improved physical processes,

increased grid resolution and improved input datasets. For instance, (Muis et al., 2017) produced a global reanalysis of storm surges and extreme sea levels (GTSR dataset) from GTSM and estimated that between 160 million people are exposed to a 1 in a 100-year flood in 2010. With the improvement of the GTSM in version 3.0, a new dataset called the Coastal Dataset for the Evaluation of Climate Impact (CoDEC) is developed and evaluated as the successor of GTSR (Muis et al., 2020). We will show that a significant part of the remaining uncertainty is caused by uncertainties in the model parameters, such as bathymetry and friction parameters.

This parameter uncertainty can be reduced by parameter estimation. In our view, this is a form of data assimilation (Heemink et al., 2002; Zijl et al., 2013; Mayo et al., 2014). Stammer et al. (2014) reported that assimilated tide models have higher accuracy than non-assimilative models. For example, Wang et al. (2021b) developed an efficient iterative parameter estimation scheme to estimate bathymetry corrections globally for a high-resolution GTSMv3.0 and significantly improved model performance in the deep ocean but improved the performance in shallow water only slightly. To further improve the model accuracy near the coastal regions, we propose a combined the estimation of bathymetry and bottom friction coefficient using more observations in the coastal areas. Bottom friction plays an essential role at the coasts accounting for the majority proportion of tide energy dissipation (Egbert and Ray, 2001). The total amount of global tidal energy dissipation is approximately 3.7TW, and two-thirds is generated by the bed stress. The bottom friction term is often modelled in the quadratic bed stress formula. There are several commonly used parameterizations, such as Chézy, Manning or White-Colebrook (Manning et al., 1890; Colebrook et al., 1937). The coefficient is often tuned using some model tests that try to reduce the difference between model and measurements. This value is difficult to set accurately but strongly related to the water level representation in the shallow water. Moreover, the bottom friction coefficient can vary strongly between regions.

In regional tide models, data assimilation is applied predominantly to estimate bathymetry, bottom friction and boundary variables (Navon, 1998; Edwards et al., 2015) with ensemble (Siripatana et al., 2018; Slivinski et al., 2017) or adjoint methods (Zhang et al., 2020). Ullman and Wilson (1998) estimated a drag coefficient by assimilating Acoustic Doppler Current Profiler (ADCP) data into a tidal model of the lower Hudson estuary with the adjoint method. Zijl et al. (2013) improved the water level forecast for the Northwest European Shelf and the North Sea through directly modeling and assimilating altimeter and tide gauge data to adjust bathymetry and Manning's roughness coefficient. Mayo et al. (2014) estimated a spatially varying Manning coefficient of an Advanced Circulation (ADCIRC) model of Galveston Bay with a square root ensemble Kalman filter. The estimation of bottom friction using data-assimilation has been applied successfully to the European Continental Shelf (Heemink et al., 2002), Bohai, Yellow, and East China Seas (Wang et al., 2021a). We found only one application at a global scale (Lyard et al., 2021), where altimetry derived tides and tides derived from tide gauges are assimilated into a combination of a time-stepping and a spectral tide model. The uncertainty for the model, is partly based on parameter uncertainty, such as bed friction, but the result is in the form of a tide dataset, called FES2014. The application focuses on tides only. We follow a different approach using time series as the basis for the cost function and also study the sensitivity of parameters and observations.

There are considerable differences between data-assimilation for tides in deep water and near the coast. In the deep ocean, bathymetry is reported as the parameter that has most influence on the tide representation (Wang et al., 2021b). The sensitivity

to bottom friction is very small in deep water, but is often the most sensitive parameter in shallow water. The main reason for this is that the effects of both parameters interact. In this study, we combined the estimation of bathymetry and bottom friction. Bathymetry directly controls the tide propagation speed, which is proportional to the square root of the local water depth (Pugh and Woodworth, 2014). On the other hand, bottom friction controls the dissipation of tide energy (Egbert and Ray, 2001). But bottom friction also decreases with depth, which results in a non-linear interaction.

Moreover, due to the quadratic velocity in the friction term, the effect of friction is enhanced when the different tidal constituents propagate along the shallow water with the complex topography (Cai et al., 2018). Thus, water level is influenced by the co-action of bathymetry and bottom friction. This also creates an interaction between the deep ocean and the shelf. The bathymetry in the deep ocean not only affects the tidal propagation there but also in adjacent coasts. And though the dissipation by bottom friction predominantly occurs in shallow water, this will also change the tides in the adjacent deep ocean. To our knowledge, this is the first study with a global model that combined the estimation of bathymetry and bottom friction. With this approach, we aim to improve modeled tides, both in the deep ocean as well as along the coasts.

We use the computation and memory efficient parameter estimation schemes proposed in our previous study (Wang et al., 2021b). Bathymetry is estimated in all ocean basins, and regions with significant tidal energy dissipation are selected for the bottom friction coefficient estimation.

The areas with most tidal energy dissipation are the European Shelf and Hudson Bay region (Egbert and Ray, 2001). FES2014 time-series are used as observations for the deep ocean. This dataset has higher accuracy for tides than our initial model (Stammer et al., 2014; Wang et al., 2021b) and with FES2014 tide time-series are generated easily for arbitrary locations and periods. In the coastal areas, tide gauge data is included in the estimation and validation processes to increase coverage for the coastal regions. Tide gauge data was collected from the UHSLC (global coverage), CMEMS (Europe). For one comparison in the Arctic, we made use of the tide dataset by Kowalik and Proshutinsky (1994), the contains four major constituents for a relatively large number of Arctic tide gauges. Since tide gauge data are scarce in some areas, we investigate the use of FES2014 also in some coastal regions. The bottom friction is estimated in the Hudson Bay region, European Shelf and often other regions with large energy dissipation using a combination of FES2014 and tide gauge data as observations. Together, these datasets form a reliable joined parameter estimation application to correct the bathymetry globally and bottom friction coefficient in the coastal and shelf seas.

In Section 2, the Global Tide and Surge Model (GTSM) and the parameter estimation scheme is introduced. Section 3 describes the strategies for the bottom friction coefficient subdomain specification and the selection of observations. Section 4 presents the parameter estimation experiment set-up and results analysis. The estimated model is evaluated with a one-year long comparison with the FES2014 dataset and tide gauge data, both in the time and frequency domains in Section 4. Finally, the discussion and conclusions follow in Section 5.

## 2 Method

### 2.1 Global Tide and Surge Model

We use version 4.1 of the Global Tide and Surge Model. It is a depth-averaged hydrodynamic model developed in the Delft3D Flexible Mesh with an unstructured grid (Verlaan et al., 2015; Kuhlmann et al., 2011). The model is forced by the tide generating potential with a full set of tide frequencies. GTSM is a combined tide and surge model to study some events such as the effect of tropical cyclones and sea-level changes on a global scale. Surge is induced by the gradients in the atmospheric surface pressure and the momentum transfer from the wind to the water.

The bathymetry used in GTSMv4.1 is a combination of the General Bathymetric Chart of the Ocean with a 15-arc second resolution globally (GEBCO 2019) and EMODnet2018 at 250m resolution in the European Shelf. GTSM has 4.9 million grid cells with a 25km resolution in the open ocean and 2.5km in the coastal zone (1.25km in Europe). We also make use of a coarser grid version of GTSM (GTSM with the fine grid and GTSM with the coarse grid hereafter). GTSM with the coarse grid has grid cells of 50km in the deep ocean and 5km for the shallow waters, resulting in 2 million grid cells. Higher resolution results in better representation of water levels but longer computation times. The CPU time used by GTSM with the coarse grid is one-third of the fine grid. The coarse grid model is used in the estimation process to reducing the computational cost with the coarse-to-fine strategy. It will be described in more detail in Section 2.2.

Tidal energy dissipation, with a total value of approximate 3.7TW, is determined by the bottom friction and internal tide frictions. Two-thirds of it, 2.39TW in GTSM, is generated by bottom friction. GTSM uses a quadratic formulation of velocity and the bottom friction known as the Chézy formula:

$$\tau_b = -\frac{\rho g}{C^2}||\boldsymbol{u}||\boldsymbol{u} \tag{1}$$

where $\rho$ is the density of water, $\boldsymbol{u}$ represents the depth-averaged horizontal velocity vector. In the Chézy formulation, C is the constant coefficient with the value of $\mathrm{C}{=}62.5(\mathrm{m}^{1/2}\mathrm{s}^{-1})$. It is important for hydrodynamic conditions. When the tide propagates over steep topographies, energy is also dissipated by generation the internal tides, contributing to a total value of 1.12TW in the abyssal depth. Internal tide friction is parameterized in the formula of the Nycander (2005) tensor scheme.

In comparison to GTSMv3.0 that was used in our previous study (Wang et al., 2021b), GTSMv4.1 contains an updated internal tide friction term that is corrected for the layer thickness in the salinity/temperature dataset. GTSMv4.1 is retweaked for the bottom friction and internal tide friction drag coefficients. GTSMv4.1 uses a full set of 484 tide potential frequencies compared to 60 constituents in GTSMv3.0 in our previous study (Wang et al., 2021b). These changes result in a more accurate initial model with the Root mean square error (RMSE) reduced by 1cm compared to the FES2014 dataset. The bias difference is removed before the RMSE calculation in this paper. In our estimation experiments, GTSMv4.1 is simulated for tide representation only and the long-term tidal constituents (SA and SSA) are excluded to avoid the seasonal changes to the time series because long-term constituents show large variation between years.

**Table 1.** Global tide models classification and resolution

| Type | Model | Resolution | Parameter estimation algorithm |
|---|---|---|---|
| Empirical tide models | GOT4.8 (Ray, 2013) | 1/2° | N/A |
| | DTU16 (Cheng and Andersen, 2017) | 1/16° | |
| | EOT20 (Hart-Davis et al., 2021) | 1/8° | |
| Pure hydrodynamic models | HYCOM (Arbic et al., 2010) | 1/12.5° | |
| | HIM (Arbic et al., 2008) | 1/8° | |
| | STORMTIDE (Müller et al., 2012) | 1/10° | |
| | ADCIRCv55 (Pringle et al., 2021) | Unstructured mesh varying between 1/4°and 1/80° | |
| Hydrodynamic model with data assimilation | FES2014 (Lyard et al., 2021) | Dataset: 1/16° | SpEnOI |
| | HAMTIDE (Taguchi et al., 2013)) | 1/8° | Variational method |
| | TPOX09 (Egbert and Erofeeva, 2002) | 1/30° | Representer-based variational method |
| | GTSMV4.1 | 1/4°in the open ocean, 1/40°in the coastal zone, (1/90°in Europe). | Dud |

### 2.2 Parameter Estimation Scheme

Global tide and surge models can be classified into three groups, empirical tide models, purely hydrodynamic models and models with data assimilation, as shown in Table 1. Some parameter estimation algorithms have been applied to global tide models. FES2014 use the Spectral Ensemble Optimal Interpolation (SpEnOI) algorithm to estimate the bottom friction coefficient, the internal tide drag coefficient, the bathymetry and the SLA. It leads to an accurate data collection of 34 tidal components. HAMTIDE is a time stepping high-resolution tide model corrected by the variational data assimilation algorithm. TPOX09 is a spectral barotropic tide model which assimilated using a variational method. However, the spectral tide model cannot describe the interaction between different tide components in shallow waters.

In this study, we use the parameter estimation scheme developed in our previous study (Wang et al., 2021b). The basic algorithm is called DUD (Doesn't use derivatives) in the generic data assimilation toolbox OpenDA (Ralston and Jennrich, 1978; ope, 2016). DUD is a Gauss-Newton-like algorithm but derivative-free to solve the non-linear least squares problems. The cost function between the model output and observations is iteratively reduced with the analyzed parameters. To estimate the high-resolution global model in an efficient way for computational cost and memory usage, as well as improving estimation accuracy, three implementations were proposed based on this algorithm in our previous study (Wang et al., 2021b):

- Computational cost reduction: Coarse-to-fine strategy

A coarse-to-fine strategy with the Coarse Incremental Calibration approach is used in the estimation process. It replaces the increments between the output from the initial model and the model with modified parameters using a coarser grid, as the equation:

$$H_f(x) \approx H_f(x_b) + (H_c(x) - H_c(x_b)) \quad (2)$$

where $\mathrm{H_c}, \mathrm{H_f}$ are the model output from the coarse and fine grid GTSM, $\mathrm{x_b}$ is the initial parameter set, and $\mathrm{x}$ is the adjusted parameter set in each analysis step. Thus, the fine model only simulates with the initial parameter set and is replaced by a coarse model in the iterations, leading to a reduction of 70 % CPU time for each model run.

– Memory requirement reduction: POD based time pattern order reduction

Parameter estimation benefits from a long simulation time, but the dimension of model output for all the ensembles also increases with longer time series. Model order reduction is a valuable technique to reduce the high dimension system with a smaller linear subspace. We project on the empirical time patterns to reduce the model output time series to a much smaller dimension. It has the advantage that the simulation length is not restricted by the Rayleigh criterion, which normally requires yearly tide simulation. As a result, the memory requirement is reduced by an order of magnitude in the parameter estimation procedure with negligible accuracy loss.

– Outer-loop iteration for nonlinear parameter estimation

Since a coarse grid model is used for the estimation iteration, we developed an outer-loop, similar to the Incremental 4D-Var described by Trémolet (2007). The inner-loop optimizes parameters using the coarse grid GTSM with the DUD algorithm. The outer-loop updates the initial model output from the fine grid model with the optimized parameters and restarts the next inner-loop. The application of outer-loop can improve the calibration performance for this non-linear model or approximate linearization.

By applying these three implementations, the parameters in GTSM can be estimated in a computation-efficient and low-memory used manner and the estimation results in a higher accuracy of tide forecast. In this approach the assimilation output is fully consistent with a forward model run that uses the estimated parameters. This allows for the use of these estimated parameters in other set-ups of the model, for example including surge or sea level rise.

### 2.3 Multiple-Parameters Estimation

#### 2.3.1 Parameters to Estimate

An estimation for a global tide model must consider the parameters in the deep ocean and shallow waters together. In the deep ocean, bathymetry and internal tide friction are two parameters affecting the model performance. Seafloor bathymetry

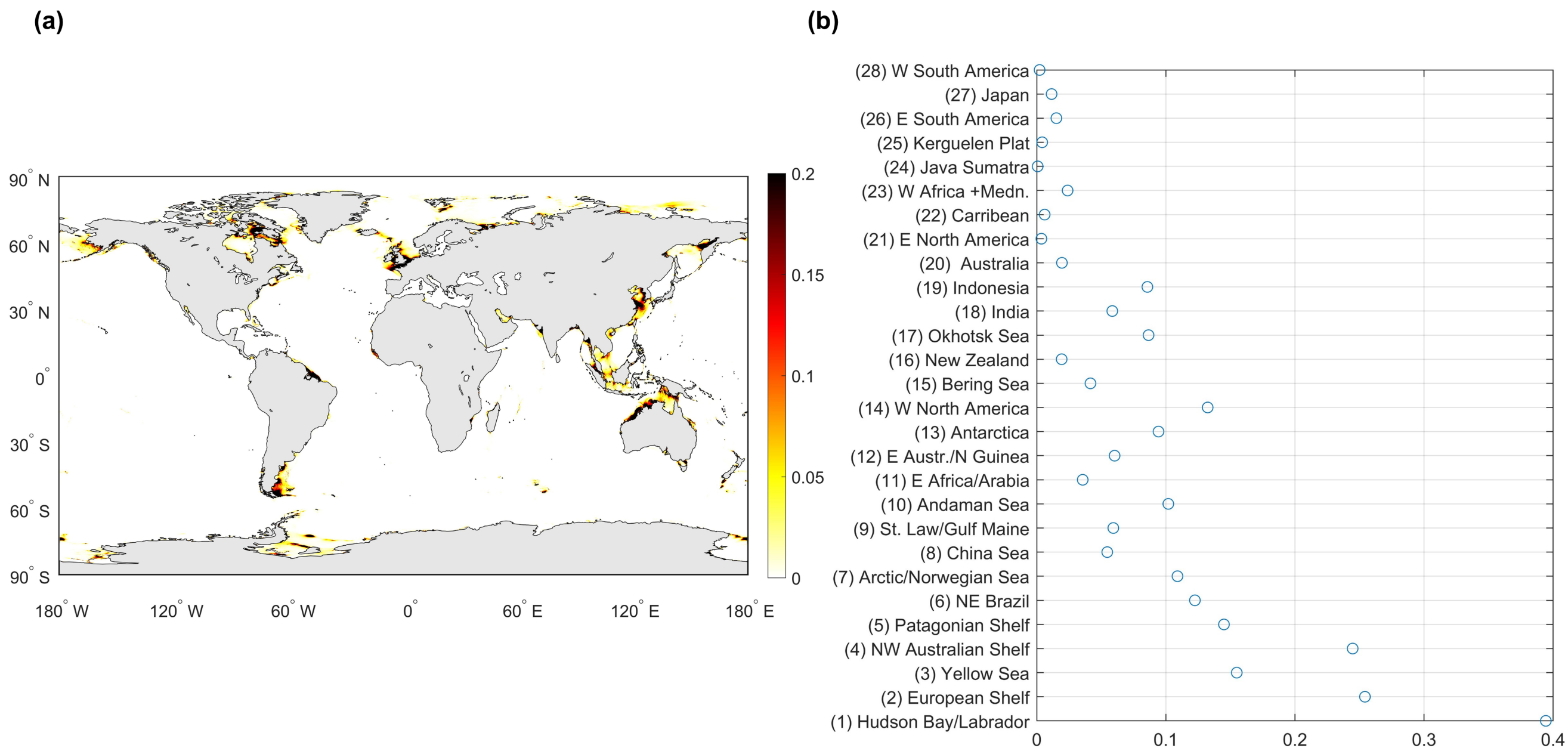

**Figure 1.** Bottom friction energy dissipation in initial GTSMv4.1 (a) Global distribution [Unit:$\mathrm{W/m^2}$]; (b) Area-integrated energy dissipation [Unit:$\mathrm{TW}$].

is of fundamental importance in many aspects of the earth, such as affecting ocean circulation and mixing. However, large parts of the global oceans remain unsurveyed. For example, Wölfl et al. (2019) reported that only about 15 % of global bathymetry datasets are based on actual data. Thus, it creates significant uncertainties and affects the sea level simulation. Internal tidal friction is a term related to tide energy dissipation in deep oceans, especially generated in the areas such as mid-ocean ridges with steep bathymetry changes. In our previous study (Wang et al., 2021b), we tested the sensitivity of bathymetry and internal tide friction term for the deep ocean by comparing the relative changes of the cost function when perturbing a specific parameter. It shows that bathymetry perturbation results in larger changes to water level than the internal tide friction term. Therefore, we only optimize the global bathymetry for the deep ocean.

In shallow water, bottom friction is also a main energy dissipative process. Figure 1a illustrates the global tide energy dissipation distribution by bottom friction term. The regions in Figure 1b are defined the same as in Egbert and Ray (2001). The total tide energy dissipation in the initial GTSM is 3.77TW, 2.39TW from bottom friction, and 1.37TW from the internal tide friction. The top values are for the Hudson Bay, the North West Australian Shelf and the European Shelf, as Figure 1b shows. We propose to estimate bottom friction only in the shallow water regions with large bottom friction energy dissipation.

It is impractical to estimate the bathymetry and the bottom friction coefficient for all the grid cells because of the limited observations and it would also computational demand, and memory requirement. To reduce the parameter dimension, we divide the global ocean into 110 subdomains for bathymetry estimation and define the correction factor for each subdomain

to adjust the parameters (Wang et al., 2021b). The detailed subdomain distribution will be shown in Section 4. The estimation subdomains for the bottom friction term are located in areas with high dissipation based on Figure 1b and sufficient coastal observations, as explained in more detail in Section 3.

#### 2.3.2 Observation Network

Global tide data from the FES2014 dataset and several global or regional tide gauge datasets were collected as observations in the calibration process.

- FES2014 dataset

  The FES2014 dataset contains 34 tidal constituents from the FES (Finite Element Solution) tide model that assimilates altimeter time series and tide gauge data (Carrere et al., 2013; Lyard et al., 2021). FES2014 data has higher accuracy than GTSMv4.1 in the deep ocean when compared with the Deep-Ocean Bottom Pressure Recorder data (Wang et al., 2021b). Moreover, FES2014 data is distributed on a regular 1/16°grid and time-series can be derived at arbitrary locations globally. Therefore, the dataset is selected to use as observations for the deep ocean to estimate bathymetry correction.

- Tide gauge data

  - UHSLC dataset: UHSLC (University of Hawaii Sea Level Center) dataset (Caldwell, 2010) contains water levels from 500 globally distributed tide gauges. The number of available locations varies in time. Stations in the UHSLC dataset are irregularly distributed, and most of the gauges are in coastal regions. We use the research quality controlled dataset, considered science-ready data.

  - CMEMS dataset: CMEMS (Copernicus Marine Environment Monitoring Service) dataset has a collection of in-situ tide gauges located in the Arctic Ocean, Baltic Sea, European North-West Shelf Seas, Iberian-Biscay-Ireland regional seas, Mediterranean Sea, and Black Sea. All the available data are published after data acquisition, data quality control, product validation, and product distribution. CMEMS data contains data for the European Shelf and is suitable for local bottom friction coefficient estimation.

  - Arctic tide gauge data with four major constituents: Kowalik and Proshutinsky (1994) described approximate tide stations in the Arctic Ocean and studied the tide performance. Four major tidal constituents, semidiurnal constituents M2 and S2, and diurnal constituents K1 and O1 are available. Since only four major tidal constituents cannot fully represent the tide time-series for calibration, they are used for the model validation to evaluate the model performance in the frequency domain.

In this study, GTSM is simulated to calibrate tides only. Firstly, we generate about 4000 time series from the FES2014 dataset to ensure enough observations for estimating bathymetry. These observations are evenly distributed and located in the deep ocean with a depth larger than 200m. Moreover, tide analysis is performed on the CMEMS and UHSLC tide gauge data on the year 2014 with the TIDEGUI software, a matlab implementation of Schureman (1958) and visual inspected of tide and surge representations. After the analysis and quality control, we obtained 237 locations in the UHSLC dataset and 297 locations from the CMEMS dataset.

## 3 Estimation of Bottom Friction Coefficient

Even though we obtained three collections of tide gauges, the observation is still quite sparse in some coastal seas. Therefore, we first investigate how to make use of the available data with the consideration of the model performance and parameter sensitivity.

### 3.1 Model and Observation Accuracy Analysis

To our knowledge, the FES2014 dataset is very accurate in the deep ocean (Stammer et al., 2014; Wang et al., 2021b) while along the coast, tide gauge data can be more trustworthy. However, tide gauges data are distributed irregularly. We propose to use a combination of the FES2014 dataset and tide gauge data in the shallow water. The first step is to analyze the accuracy of the FES2014 dataset and the initial GTSMv4.1 comparing with the tide gauge data.

Tide analysis is performed with the TIDEGUI software for the water level representation from GTSM in the year 2014. Root-mean-square (RMS) that describes the difference between model output and observations for tidal components is applied with the formula:

$$RMS = \sqrt{\overline{(A_m cos(\omega t - \phi_m) - A_o cos(\omega t - \phi_o)]^2}} \quad (3)$$

$\mathrm{A_m}$ and $\mathrm{A_o}$ are model output and observation amplitudes, $\phi_{\mathrm{m}}, \phi_{\mathrm{o}}$ are for the phase lag. $\omega$ is the tide frequency. The overbar shows the averaging over one full cycle of the constituent ($\omega \mathrm{t}$ varying from 0 to $2\pi$) in all locations. We also use Root-Sum-Square (RSS) to describe the Root Square Sum of RMS for the listed major tidal constituents. To facilitate comparison, we use the same formulas for RSS and RMS as in Stammer et al. (2014).

Table 2 illustrated the Root-sum-square(RSS) and RMS of eight major tide components between FES2014 and initial GTSM with the tide gauge data. The RSS is calculated for all the eight components in all locations. Comparing with the UHSLC dataset at the globe, FES2014 is more accurate than GTSM for all of the eight components, implying generally FES2014 dataset can provide better tide representation in the shallow water than GTSM. This conclusion is also supported by the comparison with the stations in the arctic ocean. Figure 2 shows the spatial distribution of RSS for each location, which shows that with a few exceptions FES2014 is more accurate.

**Table 2.** RSS and RMS of eight major tide components between FES2014 dataset, initial GTSM and tide gauge data in (cm)

| | RMS of all the locations | | | | | | | | RSS |
|---|---|---|---|---|---|---|---|---|---|
| | Q1 | O1 | P1 | K1 | N2 | M2 | S2 | K2 | |
| **UHSLC dataset** | | | | | | | | | |
| FES2014 | 0.37 | 1.79 | 0.83 | 2.49 | 2.66 | 11.75 | 3.49 | 0.97 | 12.98 |
| Initial GTSM | 0.53 | 2.43 | 1.17 | 3.51 | 3.17 | 15.12 | 5.37 | 1.59 | 17.03 |
| **CMEMS dataset** | | | | | | | | | |
| FES2014 | 0.45 | 1.05 | 0.62 | 1.14 | 3.96 | 18.55 | 6.99 | 2.28 | 20.42 |
| Initial GTSM | 0.68 | 2.17 | 0.68 | 1.55 | 3.22 | 17.99 | 4.66 | 1.70 | 19.15 |
| **Arctic Stations** | | | | | | | | | |
| FES2014 | - | 1.26 | - | 2.37 | - | 20.24 | 7.67 | - | 21.81 |
| Initial GTSM | - | 3.03 | - | 5.47 | - | 25.27 | 8.63 | - | 27.42 |

In the European Shelf, GTSM has the RSS of 19.15cm when comparing with CMEMS dataset, which is even smaller than the FES2014 dataset with the RSS of 20.42cm. This also can be observed from the RMS of the N2,M2,S2,K2 constituents. However, from the spatial distribution of RSS for each stations shown in Figure 2c,2d, FES2014 outperforms in most of the CMEMS locations but provides poor results in a few stations. These result in a larger RSS for FES2014 than GTSM. A possible reason is these tide components obtained from FES2014 is calculated by interpolating the gridded FES2014 dataset to the observation locations, resulting in some errors. GTSM has a higher resolution in the European Shelf, contributing to better results in those locations with complex bathymetry.

In general, FES2014 outperforms GTSMv4.1 in the shallow waters before calibration. Therefore, we will select FES2014 for calibration in the those areas where tide gauge stations are sparse. In the following, we use the FES2014 dataset in the deep ocean and CMEMS data in the shallow waters for the calibration. In addition, FES2014 is also included to support the shallow waters where without tide gauges. UHSLC and arctic stations are used for model validation.

### 3.2 Subdomains of Constant Bottom Friction Coefficient

The bottom friction coefficients in the regions with large tide energy dissipation (see Figure 1b) have to be estimated. We define multiple subdomains for the European Shelf and Hudson Bay/Labrador regions and single subdomains for other coastal areas shown in Figure 1b.

#### 3.2.1 Case region 1: Hudson Bay/Labrador region

The Hudson Bay/Labrador region, in the top one of the list 1b, generates about 0.39TW energy dissipation, 16.47% of the global sum. Most of the dissipation is concentrated in the Canadian archipelago, Hudson Bay, Foxe Basin, Hudson Strait, and Ungava Bat in Figure 3a. We defined three subdomains that firstly separate the Canadian archipelago outside the other areas.

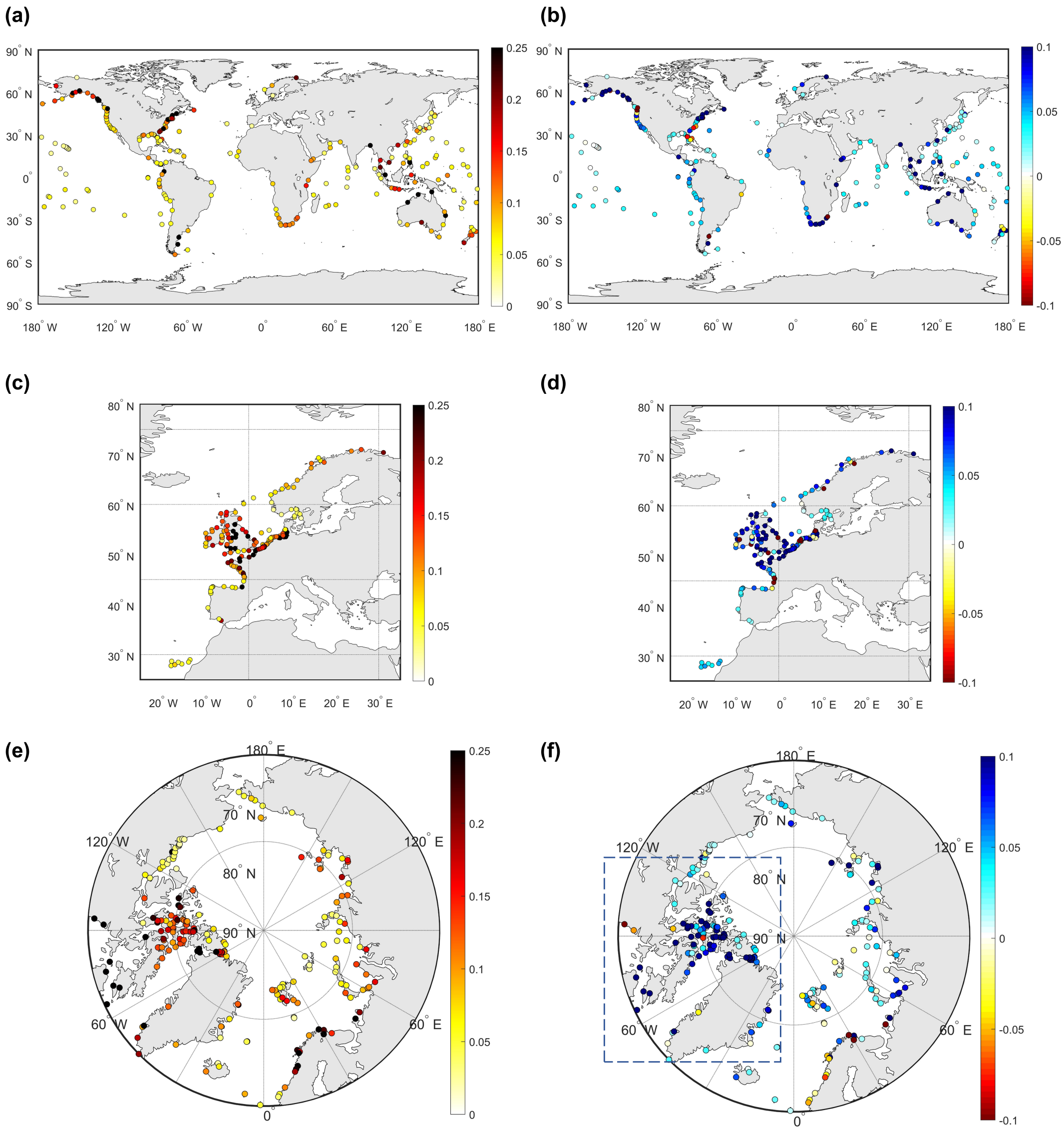

**Figure 2.** Left column: RSS between initial GTSMv4.1 and tide gauge data. Right column: RSS difference between initial GTSMv4.1 and FES2014 dataset (RSS of GTSM minus RSS of FES2014). Color blue shows better performance in FES2014 than GTSM. (a),(b) UHSLC dataset; (c),(d) CMEMS dataset; (e),(f) Arctic stations.[unit: m]

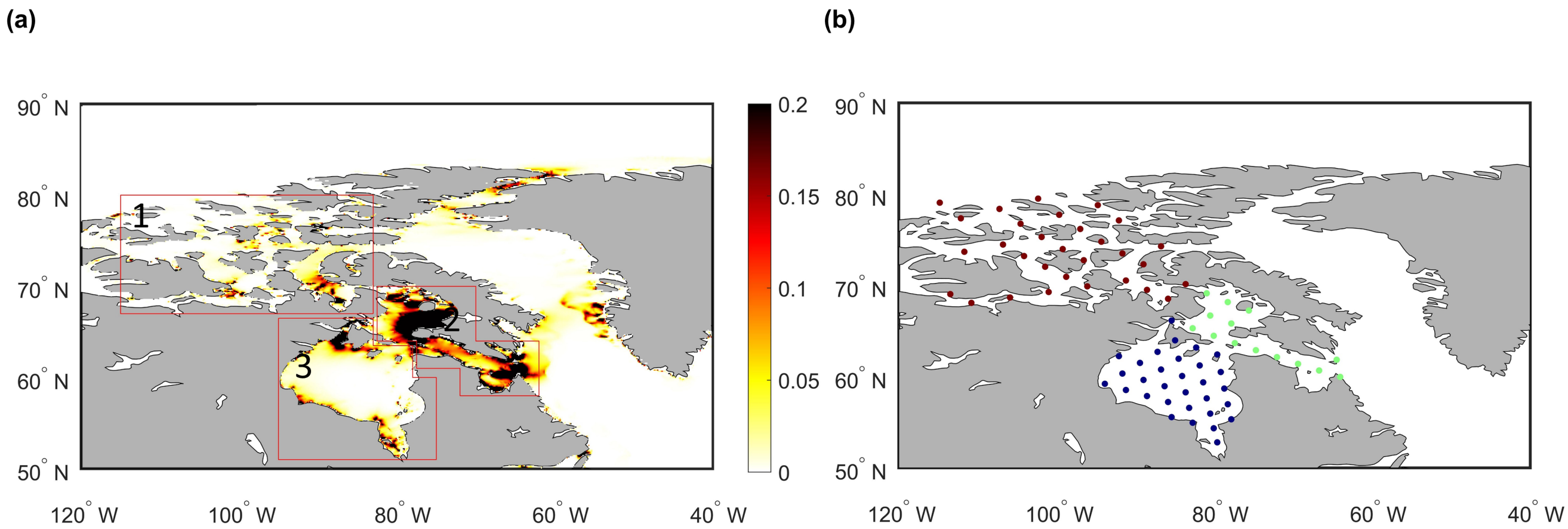

**Figure 3.** (a) Bottom friction energy dissipation per square meter of the Hudson Bay/Labrador in GTSMv4.1 [unit:$\mathrm{W/m^2}$] and bottom friction coefficient subdomains (red boxes). (b) FES2014 observation distribution: Points with different colors means in different subdomain.

Secondly, Foxe Basin, Hudson Strait, and Ungava Bay are combined as one subdomain. The last subdomain is for Hudson Bay. Subdomains are shown in the red boxes of Figure 3.

The available observations are from the arctic stations but only include four major tidal components. In theory, harmonic tide analysis can be performed for the model output and it is possible to estimate parameters with the model output in the form of tide components, but accurate tide analysis needs a time series of a year, which would increase the computation time needed for estimation by more than 10 times. In (Wang et al., 2021b), we showed that an accurate estimation can be performed with a full time series of 1 month, so this would increase run times by a factor of 12. This is not feasible for us at the moment. Therefore, we select to use the model output of time series, and these arctic stations can be utilized for the model validation.

To obtain sufficient observations, we propose to generate more observations from the FES2014 dataset because FES2014 dataset outperforms GTSM. Figure 2e illustrates the RSS (Root Sum Square) of four major tidal constituents between tide gauge data in the Arctic Ocean and the FES2014 dataset. The RSS difference between GTSM and FES2014 dataset (RSS between GTSM and tide gauge data - RSS between FES2014 and tide gauge data) varies for each location and FES2014 has smaller RSS than GTSM in most of the locations, especially in the Canadian archipelago regions (Figure 2f). The RSS of four major tidal constituents for all the locations in the FES2014 dataset is 21.81cm, while it is 27.82cm for GTSM. Errors are typically larger near the coast. Performance of FES2014 at the arctic stations is better than GTSM before the calibration. We expect the accuracy of FES in open water to be even better. Therefore, we propose to use the FES2014 dataset to as the observations in the Hudson Bay region. As a result, 61 equally distributed time series are generated as on the locations in Figure 3b. The tidal components from arctic stations can be used for the validation.

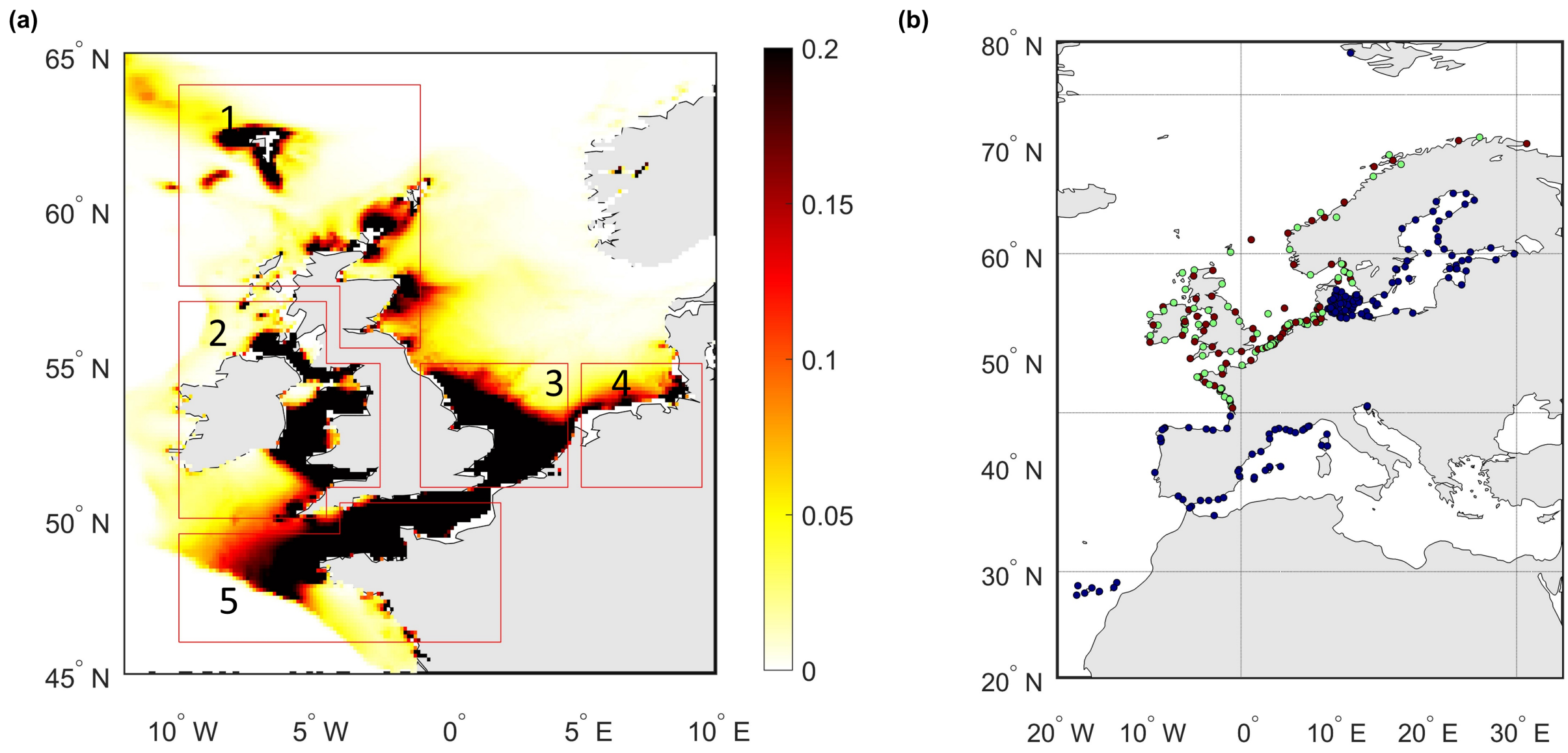

**Figure 4.** (a) Bottom friction energy dissipation per square meter across the European Shelf in GTSMv4.1 [unit: $\mathrm{W/m^2}$]and bottom friction coefficient subdomains (red boxes). (b) CMEMS observation distribution: points in red are data used for calibration, points in green are used for validation and points in blue not used.

### 3.2.2 Case region 2: European Shelf

Bottom friction energy dissipation in the European Shelf is about 0.25TW which is approximate 10.62% of the global total value shown in Figure 4. Considering the dissipation distribution, we define 5 subdomains to estimate the bottom friction coefficient. Firstly, we define subdomains for the areas in and outside the North Sea and separate the Western and Eastern part of the North Sea. Secondly,The region of Scotland, the Faro Islands and Shetland have mountainous ocean bathymetry, where expect to a higher bottom friction coefficient. The region of Scotland, the Faro Islands and Shetland have mountainous ocean bathymetry. Therefore, five subdomains are generated for the European Shelf for calibration.

The estimation for European Shelf takes advantage of a large amount of local tide gauge data (Figure 4b). About 297 tide gauge stations from CMEMS dataset are available for the year 2014, which will directly be used for parameter estimation. 132 tide gauge data stations in the Mediterranean Sea and Baltic Sea (blue points in Figure 4b) are removed because they are only weakly connected to the open ocean. The remaining stations are divided into two subsets, 70 locations for calibration (red points in Figure 4b) and 95 points for validation (green points in Figure 4b).

#### 3.2.3 Other Coastal Areas with Large Energy Dissipation

There are many other coastal regions that generate large tide energy dissipation ( 1b), like the North West of the Australian Shelf and the Yellow Sea. Therefore, 11 additional subdomains were defined globally. They are in the North West Australian Shelf, Yellow Sea, Patagonian Shelf, Okhotsk Sea, North East of Brazil, Arctic/Norwegian Sea, Antarctica, Andaman Sea, China Sea, Bering Sea, and Indonesia. Because of the limited tide gauge availability in the shallow water and limited computational resources, it is not feasible to do a detailed subdomain analysis for each of these regions. The detailed subdomain distribution is shown in the Section 4.

Time-series from the UHSLC dataset are collected at the global scale, but these measurements are not evenly distributed and lacks data in some areas, such as the North West of Brazil and the Okhotsk Sea. To make the research on these areas feasible, we propose to use more FES2014 data in these regions for estimation and use the UHSLC dataset for validation only. Therefore, additional distance-equal distributed time series were generated from FES2014 in the location with bathymetry between 50 to 200m.

In summary, we defined 110 subdomains for bathymetry and 19 subdomains for bottom friction coefficient estimation (five in the European Shelf, three in the Hudson Bay/Labrador region and 11 for other coastal areas). In total, 4061 time series from the FES2014 dataset and 70 time series from the CMEMS dataset are included in the estimation procedure. GTSM after the estimation will be validated by comparing with time-series from the FES2014 dataset in the deep ocean and tide gauge data from the CMEMS, UHSLC and Arctic stations.

## 4 Numerical Experiment and Results

### 4.1 Parameter estimation

#### 4.1.1 Experiment Design

The experiment is set up to investigate the performance of GTSM after the estimation of bathymetry and bottom friction. GTSM is simulated with tide only because no surge data is available in the deep ocean. In addition, the surge is not sensitive to the bathymetry (Wang et al., 2021b) and has to be adjusted for itself. The improvement of tide representation in this study can also benefit the accuracy of the total water level. For the estimation runs, we selected a period of one month, September 2014, which we believe is sufficient for tide calibration when using high-frequency time series with 10 minutes sampling (Wang et al., 2021b). To make this possible, meteorological and long-period signals have to be reduced as much as possible. We made model runs without atmospheric forcing and removed the SA and SSA tidal potential. These constituents were also removed from the FES2014 and tide gauge tide series to keep the comparison consistent.

The time step for the model output and observation is 10 minutes, leading to the time number in a one-month simulation equal to $\mathrm{N_t}$=4321. The number of observation locations from FES2014 and CMEMS together is $\mathrm{N_s}$=4131. Moreover, parameters are corrected for the 110 bathymetry subdomains and 19 bottom friction subdomains. In this case, the data size refers to

observation, and model output for all the ensembles (perturbed parameters) in the estimated process is about 17.3GB. With the implementation of POD-based time pattern order reduction, a truncation size of 200 represents the model output and observation in a smaller subspace of time patterns. The memory requirement is reduced by a factor of 22 after the POD application.

We defined several constraints in the optimization process to ensure that the adjusted parameters are realistic. The uncertainty for bathymetry correction factor is set to $5\%$ and for bottom friction coefficient to $20\%$. Initially, each parameter is perturbed one by one with the uncertainty value to obtain the model output for each ensemble. The same values are also used for a weak constraint adding to the cost function as the background term. It defines the difference between the initial and adjusted parameters. The background term can avoid changes to the parameter far away from the initial values than only achieve an insignificant improvement. In addition, hard constraints are also defined as the upper and lower boundary for the parameters. They are twice the uncertainty with the value of $[-10\%, 10\%]$ to bathymetry and $[-40\%, 40\%]$ to the bottom friction coefficient. Finally, there is a transition zone between each subdomain to avoid a sudden change in the correction factor from one subdomain to another. The correction factor in the transition zone is generated by automatic linear interpolation.

#### 4.1.2 Parameter estimation results

The subdomains for bathymetry and bottom friction and their sensitivity are illustrated in Figure 5. Bathymetry and bottom friction have comparable sensitivities. The sensitivity values of bathymetry vary between -0.06 to 0.02 (Figure 5a). The sensitivity of the bottom friction coefficient changes between -0.01 to 0.05 (Figure 5b), with the largest value up to approximate 0.05 in the North West of the Australian Shelf. As we discussed in the Introduction, bottom friction impacts the model performance not only in the local shallow waters but also in the nearby deep ocean. It can be observed in the Figure 5c-5f when perturbing the bottom friction in the subdomains of European Shelf and Hudson bay. RMSE is large in the nearby oceans around the perturbing subdomain and smaller when the location is far away, and the largest RMSE values are located around the Coastline (Figure 5d). Bottom friction in Hudson bay subdomain has a larger effect on the surrounding deep oceans (Figure 5c) than the European Shelf (Figure 5e). It is consistent with that the largest tide energy dissipation is in Hudson Bay.

Figure 6a illustrates the cost function changes for each iteration in these four outer loops. The first 130 iterations in each loop perturb parameter one by one; parameters are iteratively updated after that until reaching the stop criteria. Optimized parameters in this outer loop will be used as the initial parameters to start the next loop. The estimation experiment was performed with 200 cores, 9 cluster nodes, running for about 16 days, with a total cost of approximately 76800 CPU core hours.

The cost function in the experiment started from the value of $1.96 \times 10^7$. It is sharply reduced in the first outer loop to the value of $6.40 \times 10^6$, resulting in a reduction of 67.3%. The decrease of the cost function in the second to fourth outer loop is slight and converged in the fourth loop with the value of $5.58 \times 10^6$. Finally, the cost function is reduced to 28.5% of the original. The relative changes of bathymetry and bottom friction coefficient is shown in Figure 6b, 6c. After the estimation, the total tide energy dissipation is reasonable with a value of 3.77TW.

The average spatial RMSE between model output and observation in September 2014 is summarised in Table 3. Compared with the FES2014 dataset, the spatial average RMSE is sharply reduced to 47.6% after the estimation, from 5.19cm to 2.47cm.

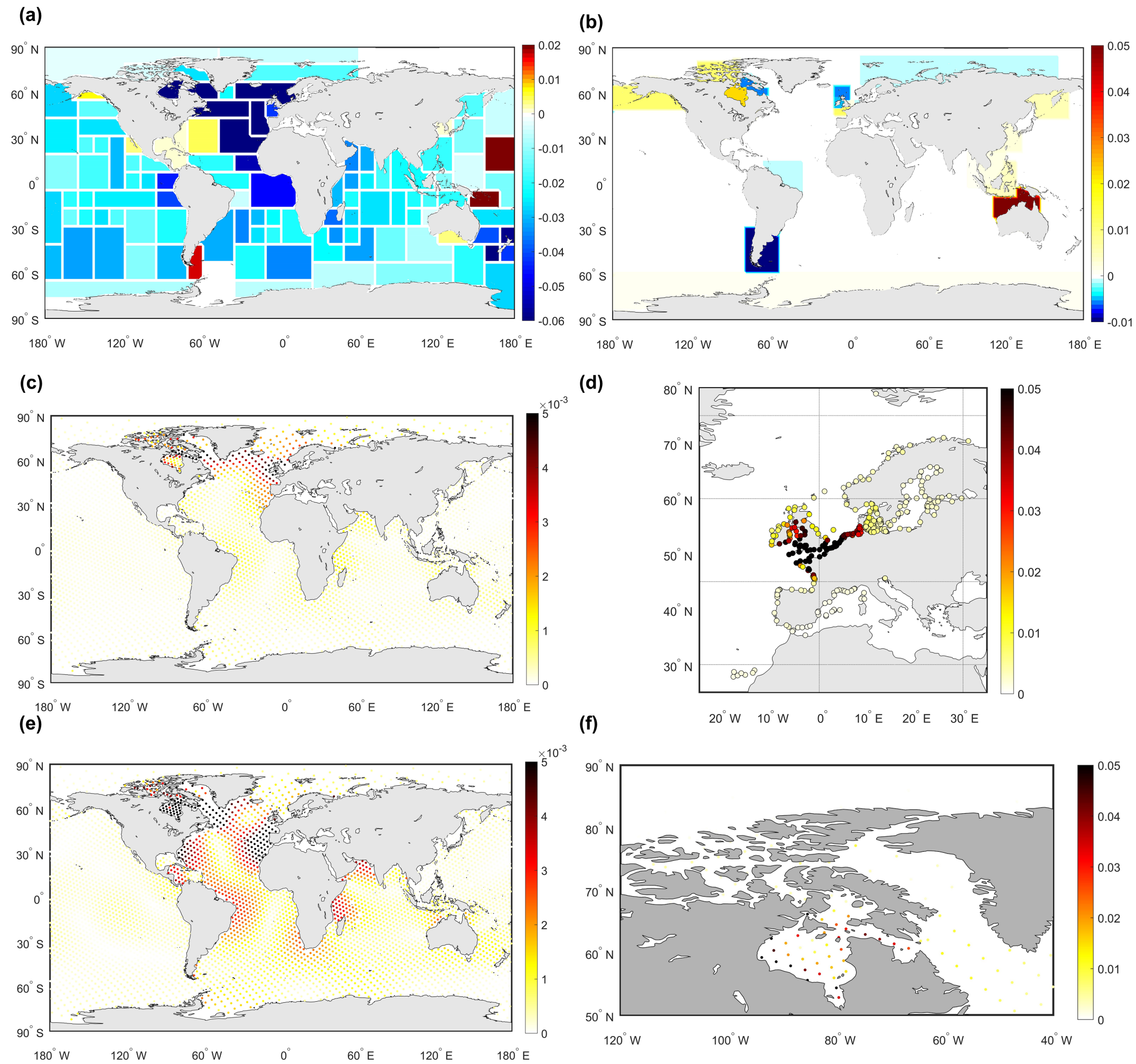

**Figure 5.** (a) Sensitivity for bathymetry. Sensitivity is the relative changes of the cost function, describing the difference between the model output and the observations when perturbing each parameter; (b) Sensitivity for bottom friction coefficient; (c-f) RMSE between initial model output and model output with perturbed bottom friction coefficient [unit: m]. (c)(d) illustrate the RMSE with the perturbation of the subdomain 5 of the European Shelf in Figure 4a; (e)(f) show the perturbation of the subdomain 3 of Hudson Bay in Figure 3a. (c)(e) show the 4061 evenly distributed locations, which are the same locations as the FES2014 dataset used in the parameter estimation. (d) shows the RMSE in the tide gauge locations around the EU. (f) shows the observation points from FES2014 dataset in detail in the Hudson Bay.

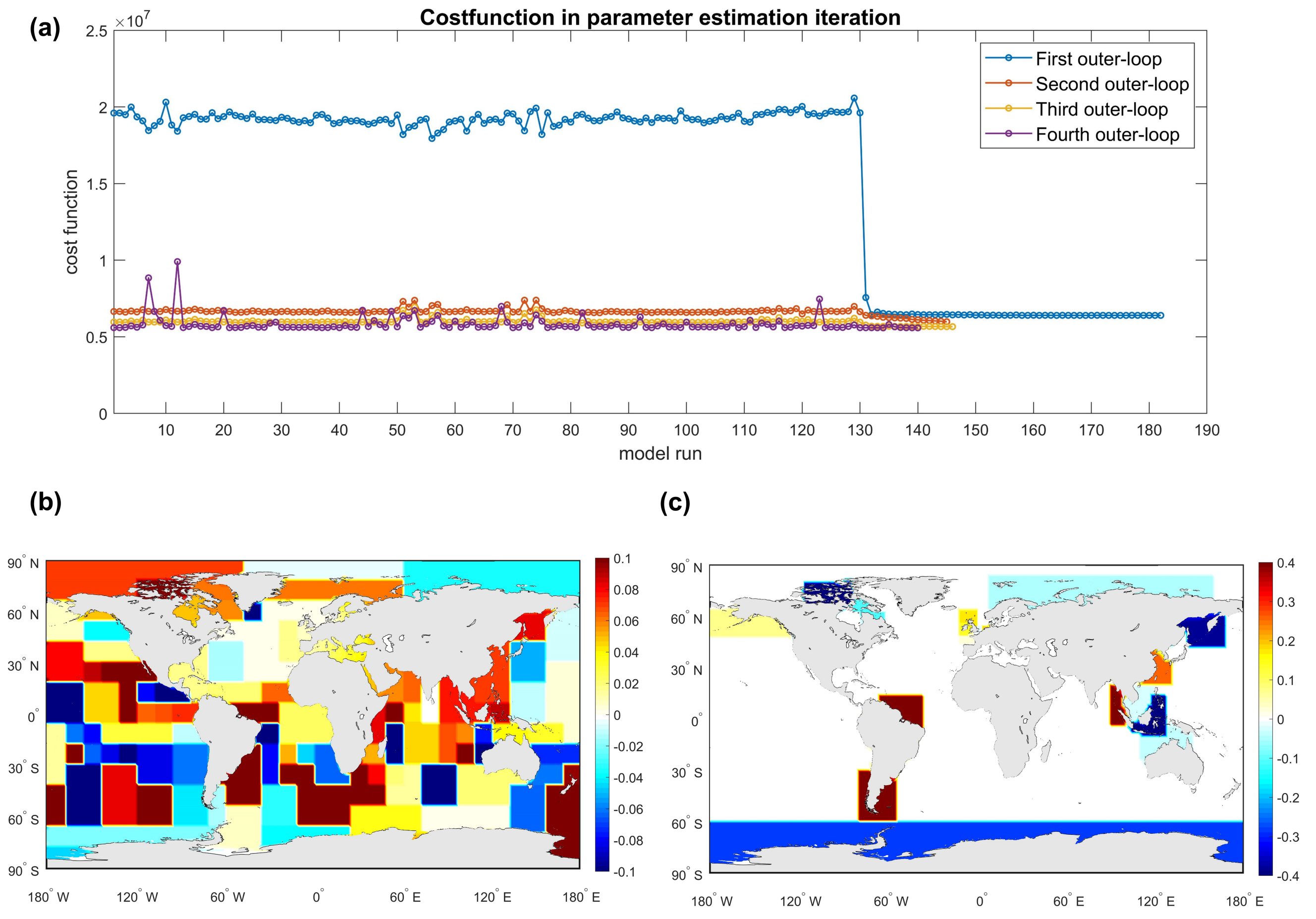

**Figure 6.** (a) Value of the cost function for each model run during the estimation; (b) Relative changes of bathymetry (bathymetry correction factor) after the estimation; (c) Relative changes of bottom friction coefficient (bottom friction coefficient correction factor) after estimation.

The total reduction is significant in the first outer loop and slight in the second to fourth outer loops. It is observed that in the Arctic Ocean, the initial RMSE with the value of 11.03cm is larger than other regions. It is observed that in the Arctic Ocean, the initial RMSE with the value of 11.03cm is larger than other regions. It is expected because we added more observation points in the Hudson Bay/Labrador. This area is more shallow with large tide amplitudes, resulting in larger RMSE than other regions. Therefore, the comparison here includes the observations located in the deep ocean and shallow water together.

The outer loop iterations provide more improvement in the Arctic Ocean than in other regions. A possible explanation is that parameter estimation impacts areas with large disagreement against observations most because they still have room to improve, and non-linear effects become more likely. In Europe, GTSM shows significant improvement compared to CMEMS tide gauge data for calibration and validation, reduced to 63.9% and 69.4%, respectively. The difference between model and UHSLC data is significantly reduced in the first outer loop and finally decreased to 75.7%. This decline is smaller than that in CMEMS data for two reasons. One is we do not include the UHSLC data in the estimation process. Secondly, many shallow waters where the UHSLC tide gauges are located are not defined for the bottom friction coefficient estimation. For example, only two tide gauges are available in the Arctic Ocean, and no stations are in the Hudson Bay area.

The spatial distribution of RMSE for estimated GTSM and the RMSE difference between the initial and estimated model in September 2014 is shown in Figure 7. RMSE between estimated model and FES2014 dataset is larger in the shallow water, such as the North West of the Australian Shelf, Hudson Bay/Labrador, than in the deep ocean (Figure 7a). It can be observed that the estimated model is significantly improved with the RMSE reduced by about 2cm for most of the regions in the deep ocean (Figure 7b).Using more time series from the FES2014 dataset in the Hudson Bay/Labrador plays a role in the estimation process since the model is excellently agreed with the FES2014 dataset for most observation points. Several locations in the middle area of Hudson Bay are a bit worse. The comparison with Arctic stations in the Hudson Bay region will be illustrated in Section 4.2.

Compared with the CMEMS dataset in Figure 7g and 7h, the parameter estimation brings a large improvement to the European Shelf, with the RMSE reduced from 17.60 cm to 11.25cm. This demonstrates that the direct use of tide gauge data in the estimation can improve model performance in shallow waters. Figure 7c and 7d also illustrates that RMSE between the model and the UHSLC dataset is decreased by a small amount. Figure 7e and 7f reports the comparison with the UHSLC dataset for the Australian shelf, where we defined several subdomains for bottom friction estimation. Even though the subdomains here are not as detailed as in the Hudson Bay and the European Shelf, the RMSE is also greatly reduced after the calibration in most of the tide gauges.

In general, GTSM after the parameter estimation showed significant improvement in September 2014. The joint estimation of bathymetry and bottom friction gives significant improvements than for estimation of bathymetry only (Wang et al., 2021b). GTSM benefits from estimating the bottom friction coefficient, especially in the Hudson Bay/Labrador and the European Shelf. The combination use of FES2014 and tide gauge data offsets the scarce supplies of observations in the shallow water and improves model skills after the parameter estimation. The direct use of tide gauge data provides excellent agreements between the observation and model output after the estimation.

**Table 3.** Average RMSE between GTSM and observations in the period of September, 2014 [unit: cm]

| Data | | Number | Initial | Es_1[a] | Es_2 | Es_3 | Es_4 |
|---|---|---|---|---|---|---|---|
| FES2014 | Arctic Ocean | 196 | 11.03 | 6.85 | 6.09 | 5.61 | 5.61 |
| | Indian Ocean | 784 | 5.31 | 2.45 | 2.38 | 2.39 | 2.38 |
| | North Atlantic | 437 | 4.89 | 2.46 | 2.40 | 2.34 | 2.38 |
| | South Atlantic | 472 | 3.75 | 2.49 | 2.33 | 2.17 | 2.16 |
| | North Pacific | 923 | 5.05 | 2.74 | 2.61 | 2.55 | 2.53 |
| | South Pacific | 1008 | 4.96 | 2.18 | 2.06 | 2.01 | 2.01 |
| | Southern Ocean | 241 | 4.96 | 3.05 | 2.87 | 2.72 | 2.65 |
| | Total | 4061 | 5.19 | 2.70 | 2.56 | 2.48 | 2.47 |
| CMEMS for calibration | | 70 | 17.60 | 12.77 | 12.15 | 11.36 | 11.25 |
| CMEMS for validation | | 90 | 16.06 | 12.47 | 11.89 | 11.21 | 11.15 |
| UHSLC dataset | Arctic Ocean | 2 | 13.18 | 9.19 | 8.34 | 6.92 | 6.63 |
| | Indian Ocean | 37 | 13.94 | 10.56 | 10.45 | 10.54 | 10.53 |
| | North Atlantic | 52 | 13.96 | 11.71 | 11.64 | 11.76 | 11.68 |
| | South Atlantic | 15 | 12.22 | 9.00 | 8.73 | 8.62 | 8.67 |
| | North Pacific | 85 | 10.52 | 8.42 | 8.33 | 8.27 | 8.22 |
| | South Pacific | 43 | 8.67 | 5.80 | 5.67 | 5.62 | 5.62 |
| | Southern Ocean | - | - | - | - | - | - |
| | total | 234 | 11.98 | 9.16 | 9.07 | 9.10 | 9.07 |

[a] Es_1, Es_2, Es_3, Es_4 means estiamted GTSM in the $1^{st}$, $2^{nd}$, $3^{rd}$, $4^{th}$ outer loop.

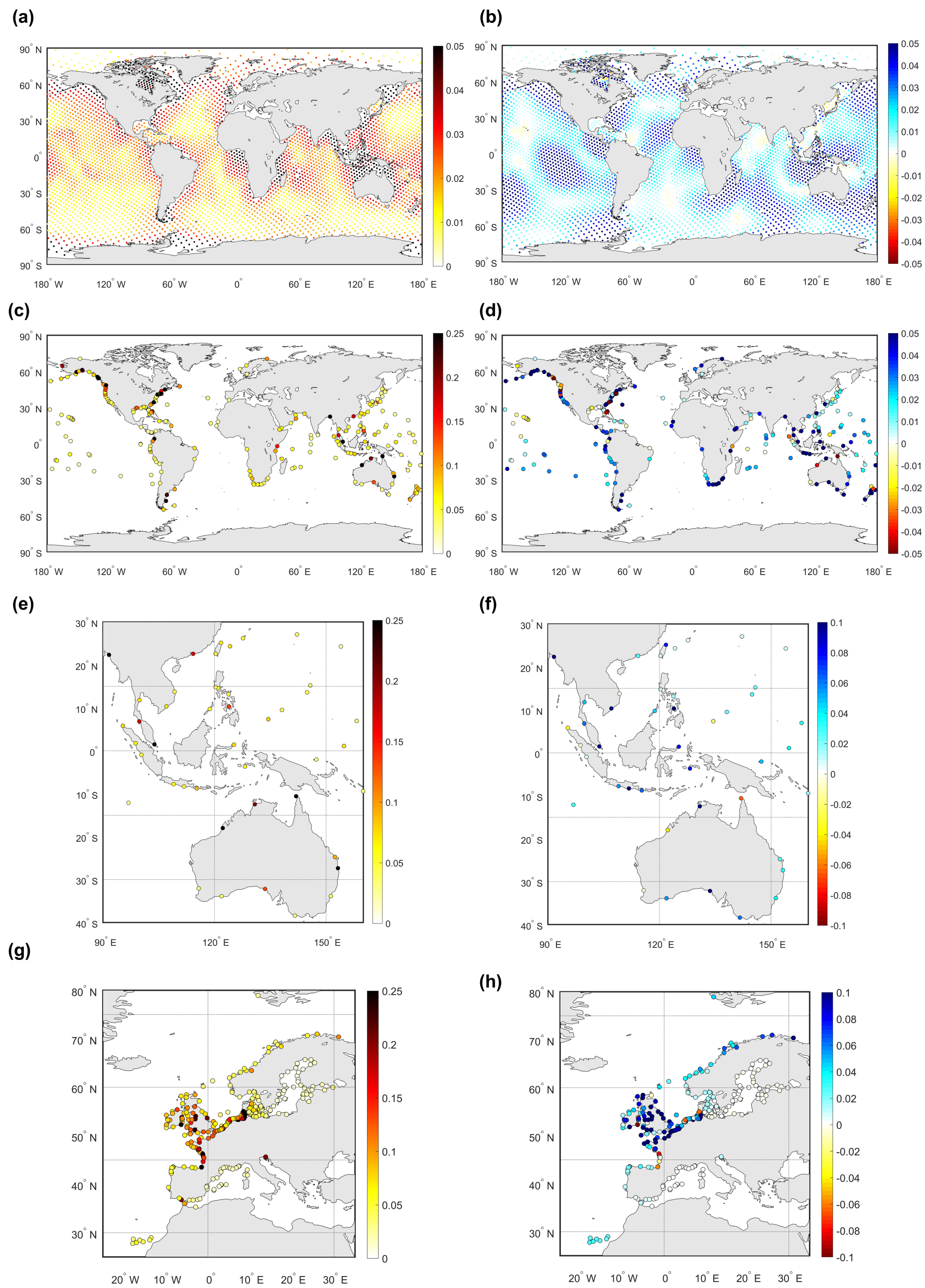

**Figure 7.** Left column: spatial distribution of RMSE of estimated GTSM. Right column: The RMSE difference between initial model and estimated model from September 1 to 30 2014. RMSE difference is defined as the RMSE of initial model minus RMSE of estimated GTSMv4.1. Color blue in Right column shows improvements in the estimated model [unit: m]. Observation dataset used to compare with GTSMv4.1:(a) (b) FES2014 dataset; (c)(d) UHSLC dataset; (e)(f) UHSLC dataset around the Australian Shelf; (g)(h) CMEMS dataset.

### 4.2 Model Validation in the Year of 2014

In this section, we validate the GTSM with the FES2014 dataset and tide gauge data for the whole year of 2014, both in the time and frequency domains.

#### 4.2.1 Monthly Time-series Comparison

First, we evaluate GTSM by comparing the tide representation of the year 2014 with observations (FES2014 and tide gauge data) in the deep ocean and shallow waters. Figure 8 shows the average RMSE between the GTSM and FES2014 time-series for each month of year 2014 in seven ocean regions (8a-8g). Most of the observations are located in the deep ocean. Compared to the initial GTSM, results in the calibration period and the other month of 2014 have reached similar accuracy, implying that the estimation is not over-fitting the observations we used. RMSE in the Arctic Ocean is larger than other regions, which coincides with the results in Table 7.

Model performance in the shallow water is compared with the CMEMS and UHSLC tide gauge data in Figure 9. CMEMS data in Figure 9a includes all the stations for calibration and validation. The average spatial RMSE for the year 2014 in the initial model is 16.7cm. After the first outer loop estimation, a large reduction is achieved to a value of 12.38cm. Accuracy is further improved due to the outer loop iteration. Finally, the RMSE is reduced to 66.5%. The direct use of CMEMS tide gauge data for calibration of bottom friction coefficient effectively reduces the model error that came from parameter uncertainty and results in high accuracy tide representation in the shallow waters.

In this study, UHSLC tide gauge dataset is only used for validation (Figure 9b). Most improvements are achieved in the first outer loop and small changes in later outer loop iterations. It shows that the calibration also has better agreements in shallow waters outside Europe. But because many of the stations are not in the estimation subdomains we defined, the improvement is limited.

To summarize, the estimated GTSM has excellent agreements with observations in the deep ocean and shallow waters compared with data in the time field. The estimation results are not over-fitting the simulation period. The direct use of tide gauge data for estimation plays a substantial role in shallow waters (here is in the European Shelf). Using the FES2014 dataset to replace tide gauges in the coastal also improves model accuracy. Model performance still can be analyzed by comparison with the tide gauges in the frequency field.

#### 4.2.2 Comparison of Tidal Constituents

To further analyze the model performance of GTSM before and after the estimation, we perform a harmonic analysis for the year of 2014.

Table 4 compares the tidal analysis results of GTSM and FES2014 before and after the estimation. Estimated GTSM has higher accuracy for all eight major tide components, with the RSS reduced to 52.3% of the original. The largest RMS is in the M2 tidal constituent. RMS of tidal constituents M2, S2, K1, and O1 in the Arctic Ocean are greatly larger than other regions before and after the estimation. This can also be observed from the spatial distribution of the amplitude and phase of

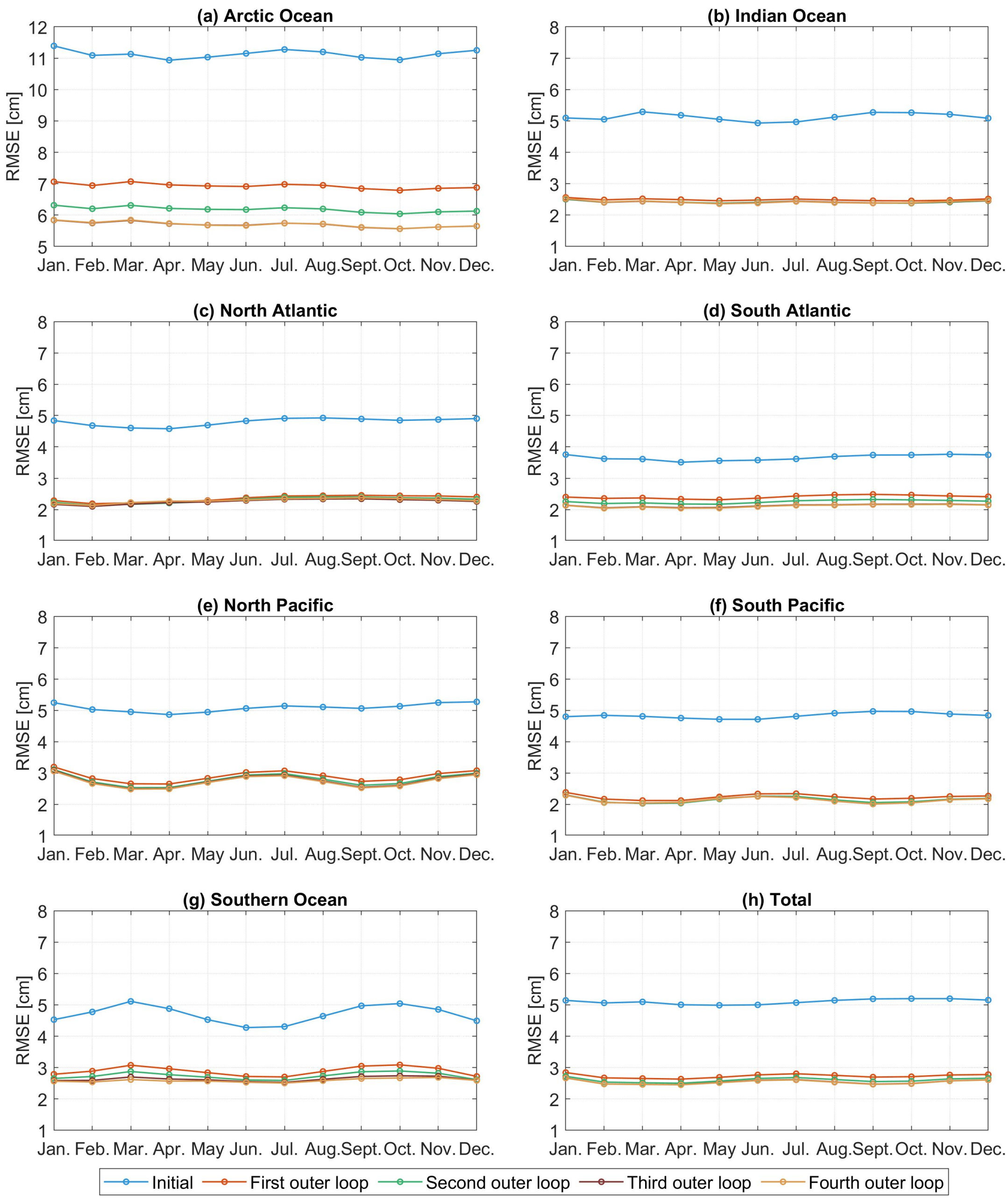

**Figure 8.** Regional averaged RMSE between GTSMv4.1 and FES2014 dataset in 2014.

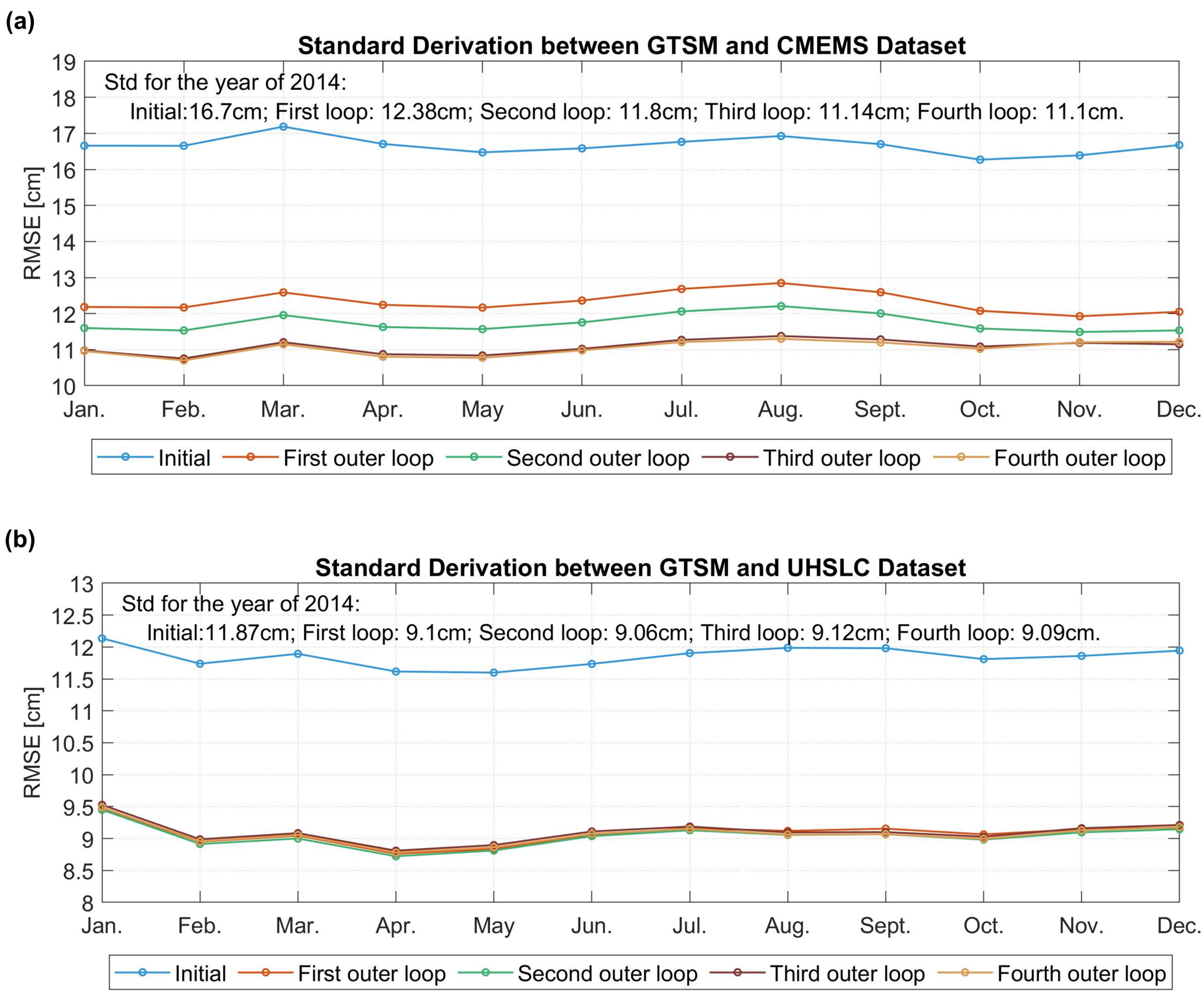

**Figure 9.** Spatial average RMSE between GTSMv4.1 and tide gauges in 2014; (a) CMEMS dataset; (b) UHSLC dataset.

the M2 tide component in Figure 10. We observed large tide amplitudes in the shallow waters, such as in the Hudson Bay, European Shelf, and the Australian Shelf, and small in the deep ocean (Figure 10a). It results in large amplitude differences between the GTSM and observations in the Hudson Bay/Labrador regions (Figure 10b), as well as the higher RMSE for the Arctic Ocean. After the estimation, amplitude and phase differences are reduced in most regions (Figure 10c, 10f). The largest amplitude differences in Figure 10c are still in the areas around the Hudson Bay, Foxe Basin, Hudson Strait, and Ungava Bay, even though the difference is significantly reduced compared with the initial model.

We compared the tide components with the Deep-Ocean Bottom Pressure Recorder (BPR) data in the deep ocean to assess the model performance with other tide models described by Stammer et al. (2014). BPR data is available from the Supplement of Ray (2013). Compared with the non-assimilative tide models, Initial GTSM has the RMS of 4.77cm in the M2 component that outperforms the purely hydrodynamic tide models described in Table 12 of Stammer et al. (2014). In the estimation process, we select the FES2014 dataset as observations for the deep ocean with a smaller RSS than the initial GTSM. FES2014 dataset is also the successor of FES2012. For example, RSS of FES2014 is 0.58cm while it is 1.12cm in FES2012 dataset

**Table 4.** RSS and RMS of eight major tide components between GTSM and FES2014 dataset in (cm). The first and second row of each region are the results before and after estimation.

| | RMS of all the locations | | | | | | | | RSS |
|---|---|---|---|---|---|---|---|---|---|
| | Q1 | O1 | P1 | K1 | N2 | M2 | S2 | K2 | |
| Arctic Ocean | 0.34 | 2.88 | 1.54 | 5.00 | 2.78 | 15.37 | 4.29 | 1.50 | 17.33 |
| | 0.27 | 1.46 | 0.72 | 2.40 | 1.69 | 8.18 | 2.67 | 0.76 | 9.27 |
| Indian Ocean | 0.24 | 0.88 | 0.62 | 1.02 | 1.16 | 5.14 | 2.32 | 0.86 | 6.01 |
| | 0.17 | 0.65 | 0.54 | 0.89 | 0.42 | 1.76 | 1.62 | 0.31 | 2.75 |
| North Atlantic | 0.25 | 0.97 | 0.40 | 1.14 | 0.81 | 4.92 | 1.45 | 0.28 | 5.44 |
| | 0.17 | 0.40 | 0.25 | 0.77 | 0.39 | 2.56 | 1.39 | 0.21 | 3.08 |
| South Atlantic | 0.26 | 0.84 | 0.42 | 1.01 | 0.84 | 3.71 | 1.72 | 0.52 | 4.44 |
| | 0.20 | 0.42 | 0.22 | 0.67 | 0.51 | 1.66 | 1.06 | 0.28 | 2.22 |
| North Pacific | 0.36 | 1.96 | 1.00 | 2.94 | 0.95 | 4.66 | 2.18 | 0.50 | 6.42 |
| | 0.29 | 1.21 | 0.76 | 2.12 | 0.52 | 1.79 | 1.34 | 0.28 | 3.46 |
| South Pacific | 0.29 | 1.16 | 0.50 | 1.27 | 0.96 | 4.22 | 2.44 | 0.57 | 5.32 |
| | 0.29 | 1.06 | 0.46 | 1.21 | 0.58 | 1.71 | 1.05 | 0.27 | 2.70 |
| Southern Ocean | 0.27 | 1.09 | 0.55 | 1.54 | 1.28 | 3.01 | 3.40 | 1.17 | 5.25 |
| | 0.24 | 0.99 | 0.48 | 1.44 | 0.95 | 1.88 | 1.13 | 0.61 | 3.07 |
| Total | 0.29 | 1.42 | 0.73 | 2.04 | 1.15 | 5.53 | 2.39 | 0.72 | 6.71 |
| | 0.25 | 0.94 | 0.54 | 1.44 | 0.64 | 2.55 | 1.40 | 0.35 | 3.51 |

**Table 5.** RSS and RMS of eight major tide components between GTSM and Deep-Ocean Bottom Pressure Recorder (BPR) data[1] in (cm)

| | RMS of all the locations | | | | | | | | RSS |
|---|---|---|---|---|---|---|---|---|---|
| | Q1 | O1 | P1 | K1 | N2 | M2 | S2 | K2 | |
| FES2012 [2] | 0.22 | 0.31 | 0.36 | 0.47 | 0.34 | 0.66 | 0.41 | 0.22 | 1.12 |
| NSWC | 0.29 | 0.87 | 0.64 | 1.29 | 1.15 | 4.27 | 1.78 | 0.66 | 5.11 |
| FES2014 | 0.14 | 0.18 | 0.14 | 0.23 | 0.19 | 0.30 | 0.27 | 0.15 | 0.58 |
| Initial | 0.29 | 1.20 | 0.55 | 1.71 | 0.98 | 4.77 | 1.97 | 0.53 | 5.71 |
| Estimated GTSM | 0.25 | 0.68 | 0.41 | 1.41 | 0.54 | 1.79 | 1.33 | 0.24 | 2.83 |

[1] BPR data is available from the Supplement of Ray (2013).

[2] Results of NSWC and FES2012 are from Stammer et al. (2014) Table 3.

435 (Table 3 in Stammer et al. (2014)). After the estimation, the RSS of GTSM is reduced to 2.83cm. Even though it is still not as accurate as FES2014 or other assimilative tide models (Table 3 in Stammer et al. (2014)), but it is excellent compared to the non-assimilative models. In addition, GTSM, like the non-assimilative models, can be used in scenario studies, such as studying climate change.

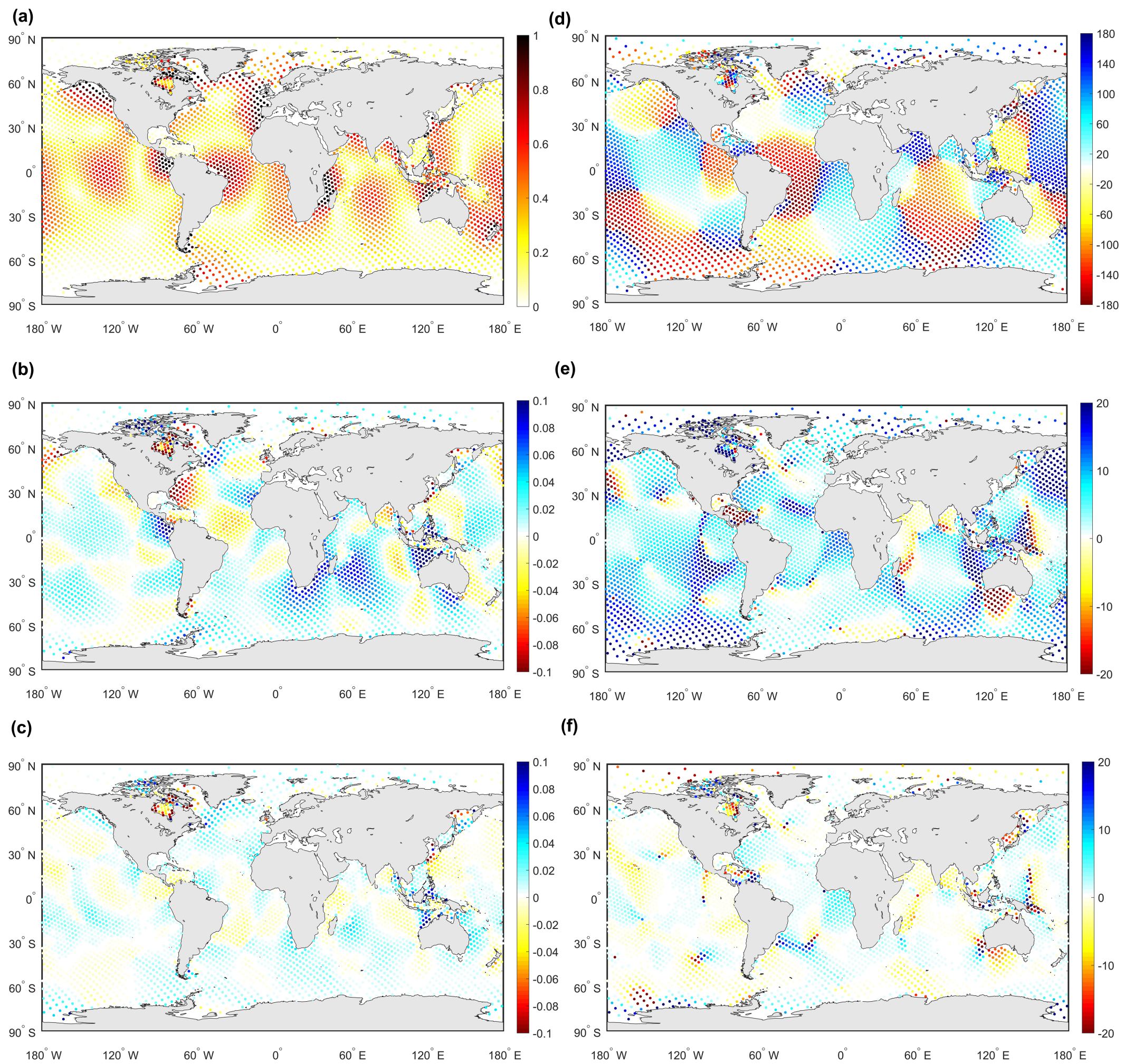

**Figure 10.** Spatial distribution of M2 amplitudes and phases from GTSMv4.1 and FES2014 dataset. (a) Amplitudes of M2 for FES2014 dataset; (b) (c): Amplitudes difference between FES2014 and initial GTSM, estimated model, respectively [unit: m]. (d) Phases of M2 for FES2014 dataset; (e) (f): Phases difference between FES2014 and initial GTSM, estimated model, respectively [unit: degree].

To analyze GTSM performance in shallow waters, we summarized the RMS of major tide components with the comparison of tide gauge data in Table 6. Tidal components from the FES2014 dataset have been evaluated in the tide gauge locations in Table 2. After the estimation, the RSS of GTSM is reduced by 16% of the initial GTSM, from 17.03cm to 14.36cm. However, the error is still larger than in the FES2014 dataset with the value of 12.98cm in Table 2. It is expected because we use the FES2014 dataset as the observation for some coastal regions, and the observation error limits the estimation accuracy to some extent.

Compared with the CMEMS dataset (all locations in the calibration and validation subsets), the RSS of all eight components is reduced from 19.15cm to 12.74cm. Moreover, after the estimation, model errors have the largest reduction in the European Shelf compared with CMEMS than other regions compared with the UHSLC dataset and arctic stations. These results also demonstrate directly assimilating tide gauge data can significantly improve the accuracy of tide representation in models.

In the Arctic Ocean, we analyze the four major tide components from arctic stations and GTSM. When comparing with the FES2014 dataset in the Arctic Ocean (Figure 8a), model error is significantly decreased in every outer-loop iteration. To assess the model performance in each iteration, we reported results with the comparison of arctic stations in four outer-loops in Table 6. RMS is reduced after the first outer loop, especially for the M2 component, resulting in the value of 22.24cm. It is close to the accuracy of the FES2014 shown in Table 2. However, the total accuracy in the second to fourth outer loop is not further improved. M2 constituent becomes a bit worse, but other tide frequencies are improved. This is contrasted with we observed from the Table 3 and Figure 8a of the comparison with FES2014 data in Arctic Ocean. In Table 3, the RMSE of 196 time series in the Arctic Ocean derived from the FES2014 dataset is reduced step by step with the implementation of outer-loop iterations. Model output is continuously close to the FES2014 dataset in this process, but there are no significant improvements to the Arctic stations from the outer-loop iteration. This is because, firstly, most of the arctic stations are located in the Canadian archipelago, not the Hudson Bay. In addition, there are still observation errors in FES2014 even though FES2014 provides higher accuracy than the initial GTSM. Estimation leads the results closer to the FES2014 but does not mean constantly closer to the Arctic Stations because of the observation error in FES2014 and the uncertainties with the arctic stations. The spatial distribution of RSS for each station is illustrated in Figure 11. We can observe that error of GTSM after estimation is smaller than before (Figure 11a-c). However, the estimated GTSM does not surpass the accuracy of the FES2014 dataset (Figure 11d), which we also did not expect. Therefore, it is concluded that the observation error significantly influences the estimation accuracy. In addition, stations in Norway seem to get worse (Figure 11c), which is inconsistent with CMEMS data.

In summary, model assessments from the time and frequency fields demonstrate that the parameter estimation of bathymetry and bottom friction coefficient combined with the FES2014 and tide gauge data as observation can significantly improve the tide representation in the deep ocean and shallow waters.

## 5 Conclusions

This study presents a study about the joint estimation of bathymetry and bottom friction coefficient, for a Global Tide and Surge Model (GTSM), which effectively improves the global tide representation, especially in shallow waters. Bathymetry is the main

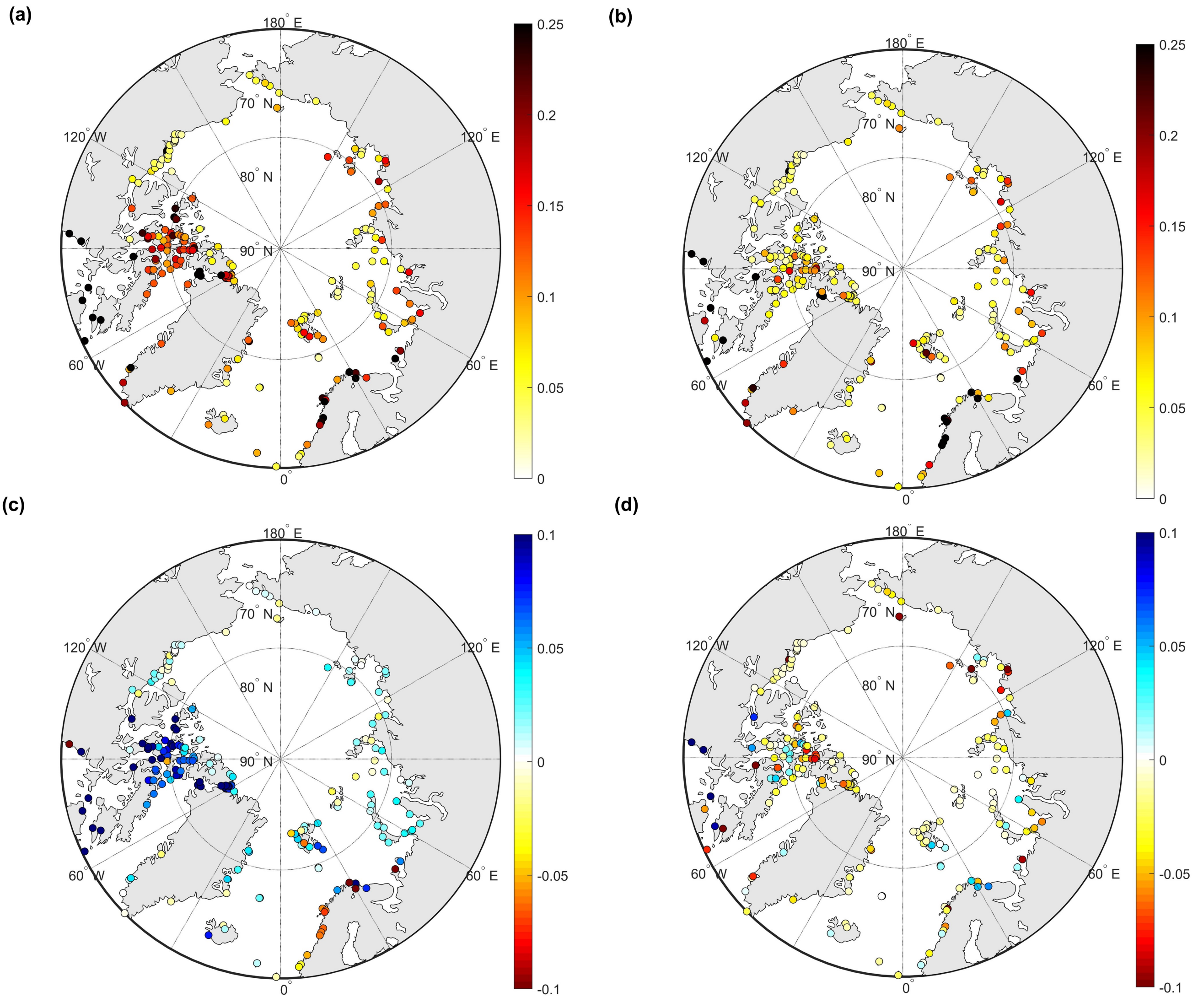

**Figure 11.** RSS of four major tide components between the Arctic station and initial GTSMv4.1 (a), estimated GTSMv4.1 (b); (c) RSS difference between initial model and estimated model (RSS of initial model minus RSS of estimated model); (d) RSS of estimated model minus RSS between FES2014 and Arctic Stations [unit: m]. (color blue shows better performance in estimated GTSM than initial model (c) or FES2014 dataset (d).

**Table 6.** RSS and RMS of eight major tide components between GTSM and CMEMS, UHSLC and Arctic Stations in (cm)

| | RMS of all the locations | | | | | | | | RSS |
|---|---|---|---|---|---|---|---|---|---|
| | Q1 | O1 | P1 | K1 | N2 | M2 | S2 | K2 | |
| **UHSLC dataset** | | | | | | | | | |
| Initial | 0.53 | 2.43 | 1.17 | 3.51 | 3.17 | 15.12 | 5.37 | 1.59 | 17.03 |
| Estimated GTSM[1] | 0.51 | 2.21 | 1.05 | 3.24 | 2.71 | 12.63 | 4.56 | 1.29 | 14.36 |
| **CMEMS dataset** | | | | | | | | | |
| Initial | 0.68 | 2.17 | 0.68 | 1.55 | 3.22 | 17.99 | 4.66 | 1.70 | 19.15 |
| Estimated GTSM | 0.51 | 0.85 | 0.57 | 1.48 | 2.45 | 11.19 | 5.12 | 1.23 | 12.74 |
| **Arctic Stations** | | | | | | | | | |
| Initial | - | 3.03 | - | 5.47 | - | 25.27 | 8.63 | - | 27.42 |
| Es_1[2] | - | 2.22 | - | 3.74 | - | 20.39 | 7.73 | - | 22.24 |
| Es_2 | - | 2.11 | - | 3.51 | - | 20.68 | 7.52 | - | 22.38 |
| Es_3 | - | 1.98 | - | 3.24 | - | 20.65 | 7.27 | - | 22.22 |
| Es_4 | - | 1.95 | - | 3.24 | - | 20.46 | 7.21 | - | 22.02 |

[1] Estimated GTSM is the estimated GTSM in the fourth outer loop.

[2] Es_1, Es_2, Es_3, Es_4 means estimated GTSM in the $1^{st}, 2^{nd}, 3^{rd}, 4^{th}$ outer loop.

parameter affecting model performance at the worldwide scale (Wang et al., 2021b), and the bottom friction term influences the tide representation in areas with significant tide energy dissipation (shallow/coastal areas). The FES2014 dataset, with higher accuracy than the initial GTSM in the deep ocean, is used for calibration in this paper. It plays a vital role in correcting the bathymetry factor in the oceans domain we defined. To ensure that the estimation for bottom friction coefficient is feasible, we propose a combination of FES2014 and tide gauge data for the estimation of bottom friction in shallower coastal waters. Applying this parameter estimation significantly improves the tide representation of GTSM almost everywhere around the globe.

The Hudson Bay/Labrador Sea and European Shelf are the regions with the largest tide energy dissipation. The bottom friction coefficient in the European Shelf is optimized with the tide gauge data from the CMEMS dataset. This results in the largest improvements of tide accuracy for shallow waters. We refined the observation locations from the FES2014 dataset in the Hudson Bay and Labrador sea. This approach is based on the condition that data of Arctic stations only have four major tide components that cannot be used for calibration, and FES2014 has higher accuracy than initial GTSM when comparing against these stations. After estimation the accuracy of GTSM is close to that of FES here. Moreover, some other coastal areas with large energy dissipation are estimated by including more observation located in the depth between 50-200m from FES2014 dataset because the numbered UHSLC tide gauges are too few to be used for calibration directly in many regions. After calibration, GTSM has smaller disagreements than initial model but not as accurate as the FES2014 dataset when comparing

with UHSLC dataset. RSS of eight tide components between FES2014 and UHSLC tide gauge data is 12.98cm, which is smaller than the estimated GTSM with the value of 14.36cm.

In summary, the accuracy of GTSM is significantly improved with the combined parameter estimation of bathymetry and bottom friction coefficient. Tide representation in shallow waters benefits from the optimization of bottom friction coefficient, contributing to a more accurate water level forecast when including wind and air pressure conditions for surge simulation. Accurate parameter estimation for global tide models needs sufficient observations and a proper determination of parameter subdomains. Direct utilization of tide gauge data provides the most significant reduction of model error. Some areas such as the Hudson Bay with insufficient tide gauge measurements, the use of other data with higher accuracy than the model can also improve the model performance to a certain extent. The one-year model validation demonstrates the estimated GTSM can provide long-term high accuracy tide forecasts. Thanks to the efforts of communities like GLOSS, UHSLC, CMEMS, EMODnet and GESLA, more and more tide gauge data is becoming available. However, the spatial scales in shallow coastal waters are much smaller than in deep water, so that the number of available tide gauge is not yet sufficient for calibration of tide models at the moment. Satellite altimetry has the potential to add much more information about tides in shallow waters. However, compound tides, overtides and tide-surge interaction will make this more complicated than in deeper waters.

*Code and data availability.* The Delft3D Flexible Mesh software can be obtained from Deltares (https://oss.deltares.nl/web/delft3dfm). The OpenDA software can be obtained from https://www.openda.org/. The FES2014 dataset is acquired from https://www.aviso.altimetry.fr/. Research Quality Data of UHSLC is made available at the University of Hawaii Sea Level Centre with the link ftp://ftp.soest.hawaii.edu/uhslc/rqds. The CMEMS data can be obtained from https://marine.copernicus.eu/. Bathymetry data is available from https://www.gebco.net/data_and_products/gridded_bathymetry_data/ (GEBCO 2019) and https://emodnet.ec.europa.eu/en/bathymetry (EMODnet).

*Author contributions.* XW and MV conceived the study and designed the parameter estimation scheme. JV prepared the GTSMv4.1. XW performed the estimation experiments and carried out the data analysis. MV, HL, and JV provided useful comments on the paper. XW prepared the manuscript with contributions from MV and all other co-authors.

*Competing interests.* The authors declare that they have no conflict of interest.

*Acknowledgements.* The first author wishes to thank the China Scholarship Council for providing financial support in terms of a scholarship grant. This work was carried out on the Dutch national e-infrastructure with the support of SURF.

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
