# Peer review of "Data-assimilation based parameter estimation of bathymetry and bottom friction coefficient to improve coastal accuracy in a global tide model"

_Ocean Science, 2021_

## Author Response (AR1)

**Response to reviewer comments**

Manuscript Number: os-2021-112

Author(s): Xiaohui Wang; Martin Verlaan; Jelmer Veenstra; Hai Xiang Lin

Paper title: Parameter Estimation to Improve Coastal Accuracy in a Global Tide Model

Dear editor:

We are submitting the revised manuscript title "Data-assimilation based parameter estimation of bathymetry and bottom friction coefficient to improve coastal accuracy in a global tide model". Title is changed from the previous name. Firstly, we would like to thank you for your precious time and invaluable comments.

We believe that the revised version of our paper addresses all your concerns and has resulted in significant improvements. The detailed response to each comment can be found in the attached document. We are convinced that our parameter estimation scheme for the high-resolution Global Tide and Surge Model has shown its ability to provide high accuracy forecast for tide and surge, which warrants the interest of the diverse audience of Ocean Science. As such, we believe that this new version is suitable for publication. We made sure that every change to the manuscript has been clearly documented below.

Sincerely,

Xiaohui Wang

On behalf of all co-authors

**RC1 Anonymous referee #1**

**General comments**

This manuscript deals with data assimilation-based parameter estimation of bathymetry and bottom friction coefficient to improve coastal accuracy in a global tide model. Ultimately, the purpose of this study is to improve tidal prediction accuracy of their GTSM through data assimilation using FES2014 and tidal gauge data. Regarding this point, I wonder if the GTSM in tidal prediction can be better than the FES2014. If not, what is advantage of use in the GTSM? Just computation and memory efficiency? In addition, with respect to the parameter estimation of bathymetry, I suggest that the authors compare their model initial bathymetry and corrected bathymetry with that of FES2014. These results may provide useful information on their input bathymetry's suitability.

In general, I do not think that the manuscript is well written because of a lot of unclear and repeated explanations. The authors should be avoid report style and should make the manuscript concise with stressing their novel scientific findings. Additionally, the location map with names should be added for readers to easily understand locations mentioned in this study. Therefore, as it is, it seems to me that this manuscript is not appropriate to publish in Ocean Science.

Response: Thanks for the review and suggestions. It seems that we have not made our motivation and choices clear enough. FES2014 is an assimilative tide model, that comes in the form of a gridded data collection with the resolution of 1/16°. It consists of 34 tidal components to provide tidal representations. This gives an accurate estimate of the tide, but the underlying model is only used as a first guess or weak constraint. As a result of this choice, the tidal solution can be very accurate, but the result is a relatively static dataset. In contrast, the calibration of GTSM in this paper uses the model as a strong constraint. This results in a calibrated model that can used as a regular non-assimilative hydrodynamic model. For example we use the GTSM model for storm-surge forecasting and studying the impact of sea level rise; both are not possible with FES2014. A consequence of using the model as a strong constraint is that this dramatically reduces the number of degrees of freedom for the assimilation, leading in general to larger differences with the observations. Comparing FES2014 to the calibration of GTSM, the aim is a different type of result. While we aim for a good accuracy, it is likely that the calibrated GTSM produce less accurate tides but can be used for a wider range of applications. These remarks discuss the assimilation aspect only, but other factors, such as the resolution, quality of the input data and the physics included in the model, also contribute to the accuracy of the final result. Finally, also the amount and quality of the assimilated observations influences the accuracy. FES2014 assimilates a large number of observations, both from remote sensing and in-situ achieving a very high accuracy in deep waters, which is why we have selected it as a data-source for our calibration.

This description to explain the difference between FES2014 and our research has been added in the Introduction of the manuscript as follows:

" Lyard et al. (2021), assimilated altimetry tides and tide gauge data into into a combination of a time-stepping and a spectral tide model. The uncertainty for the model, is partly based on parameter uncertainty, such as bed friction, but the result is in the form of 34 tidal components in a gridded data collection with the resolution of 1/16°, called FES2014. It can provide accurate estimate of tide, but the result is a relatively static dataset because the underlying T-UGO tide model is only used as a first guess or weak constraint. In contrast, we propose a different approach to calibrate GTSM in this paper, using the model as a strong constraint. This

results in a calibrated model that can be used as a regular non-assimilative hydrodynamic model. For example, we use the GTSM model for storm-surge forecasting and studying the impact of sea level rise; both are not possible with FES2014. " (Highlighted in Page 2)

We also add one paragraph to discuss the accuracy of FES2014 and the estimated GTSM in the conclusion of the manuscript.

"However, the purpose of the calibration of GTSM is different from that of FES2014. GTSM is used as a strong constraint. A consequence of it is that this dramatically reduces the number of degrees of freedom for the assimilation, leading in general to larger differences with the observations. It is likely that the calibrated GTSM produce less accurate tides but can be used for a wider range of applications. These remarks discuss the assimilation aspect only, but other factors, such as the resolution, quality of the input data and the physics included in the model, also contribute to the accuracy of the final result. Finally, also the amount and quality of the assimilated observations influences the accuracy. FES2014 assimilates a large number of observations, both from remote sensing and in-situ achieving a very high accuracy in deep waters, which is why we have selected it as a data-source for our calibration." (Highlighted in Page 31)

The bathymetry datasets used in the T-UGO tide model are GEBCO and ETOPO, but after applying data assimilation to the model, the FES2014 tidal constituents are not fully consistent with the bathymetry. Therefore, we have not attempted to compare the bathymetry. Besides, we have contacted the authors for the availability of the bathymetry data, but until now we didn't receive it.

We have carefully checked the manuscript and made a number of changes to improve the structure and readability, as listed:

1) We have checked the grammar of the manuscript. Some repeated explanations are removed and redundant descriptions to make the manuscript more concise.
2) We have emphasized the scientific findings in the section Introduction, Experiment and Conclusion.
3) The section Experiment have been rewritten. A one-year simulation experiment comparing the surge and total water level representations before and after the estimation was added into the subsection "Monthly Time-series Comparison" of the paper. Results show that even though surge simulation keeps the same accuracy after the estimation, the water level forecast accuracy is improved because of the improvement of tide representations.
4) The location map with names has been added to the Figure 3 and Figure 4 in the Manuscript. It also can be found in the point-to-point response.

Some specific comments follow to help the authors address their manuscript's weakness:

1. Title

- The authors should change the title to contain key words (e.g., Data assimilation based parameter estimation of bathymetry and bottom friction coefficient to improve coastal accuracy in a global tide model).

Response: Thank you for your suggestion. The title is changed to: Data-assimilation based parameter estimation of bathymetry and bottom friction coefficient to improve coastal accuracy in a global tide model

2. Abstract

- I think that the authors need to include the specific parameter estimation scheme name used for an efficient computation and memory efficiency.

**Response:** We now call the parameter estimation scheme: Time-POD based coarse incremental parameter estimation. This name has been adjusted in the paper.

3. Section 1 (Introduction)

- On p. 3 lines 68-69: The authors need to clearly explain how the energy dissipation by bottom friction in shallow water also change the tides in the adjacent deep ocean.

**Response:** Thank you for your suggestions. We have added the following explanation into the Introduction of the manuscript.

"Though the dissipation by bottom friction predominantly occurs in shallow water, this will also change the tides in the adjacent deep ocean when the tide propagates from the coastal regions to the nearby deep ocean. The range of affected areas are related to the topography and tide dissipation (Detailed analysis see Section 4.1)." (Highlighted in Page 3)

This is illustrated by the result of experiments with perturbed bottom friction coefficient in one coastal subdomain, as the following figure shows.

[Figure]

Figure: RMS of the difference between the initial model and model with perturbed bottom friction in the European Shelf (a) and Southern Ocean (b) in [m]. Bottom friction coefficients in the red boxes are perturbed with 20%.

We can observe that the influence of the tidal dissipation on the shelf is much wider than the shelf itself, so in the parameter estimation we now get better results by coupling the calibration of bathymetry and friction. The RMS in Figure a is obviously larger than Figure b. It is consistent with the fact that more tide energy dissipation occurred in the European Shelf than in the Southern Ocean.

4. Section 2 (Method)

- The authors should make it clear whether they adjusted the model bathymetry or not. Because GEBCO 2019 is sourced from navigation chart data, the chart datum can be not mean sea level but lowest astronomical tide (LAT) or a datum as closely equivalent to this level. Thus,

particularly in tidally dominated shallow coastal regimes, the GEBCO 2019 should be adjusted. I also recommend that the authors compare their model depth data with that of FES2014 which can be provided as request.

Response: Our GTSM model uses Mean Sea Level (MSL) as its vertical reference. To be consistent, input datasets should be converted to the same vertical reference. We applied corrections to both GEBCO and EMODnet datasets. For the GEBCO dataset, the MSL-LAT correction was calculated with the FES2012 dataset and applied everywhere. Although GEBCO does not explicitly defined the reference as LAT, our experience is that much of the data is referenced to LAT, which is consistent with the IHO standard for nautical charts.

We add a description of the reference datum in the section Method of the manuscript:

"For consistency between the vertical reference of the model and that of the data, all bathymetric data are corrected using the Mean Sea Level (MSL) as its vertical reference datum." (Highlighted in Page 4)

In addition, the question referred to the comparison of model depth data with that of FES2014 has been answered in response to the general comments.

- On p. 4 line 108: Need to put reference for Chezy formula.

Response: Chézy formula was developed by the French engineer, Antoine Chezy. We now include a reference in the paper (Manning, 1891.) (Highlighted in Page 4)

Manning, R., "On the flow of Water in Open Channels and Pipes." *Transactions Institute of Civil Engineers of Ireland, vol. 20, pp 161-209, Dublin, 1891, Supplement, vol 24, pp. 179-207, 1895*

- On p. 4 line 110: As far as I know, the value of C varies with depth range. Need to check it and clarify it.

Response: There are several types of formula to define C in the bottom friction term. We use the Chezy formulation, C is defined as the constant coefficient. For the manning friction formula, the value of C is dependent on the depth: $C = \sqrt[6]{D}/n$, where D is the depth and n is the user-defined coefficient.

- On p. 4 lines 117-118: As the authors showed in Table 1, even though the resolution of TPOX09 is higher than that of FES2014, they used FES2014 without any clear explanation. With respect to this point, they need to clearly explain the reasons. Did they calculate RMSE of TPOX09 and compared with that of FES2014?

Response: The accuracy of FES2014 and TPX09 is comparable even though TPXO9 has a higher resolution than FES2014. Stammer et al., 2014 assessed several model performances including FES2012 and TPXO8. The following table shows the comparison from the paper of Stammer et al., 2014.

| RMS Model Differences (cm) Against Deep-Ocean Bottom Pressure Recorder (BPR) Stations | | | | | | | | | | |
|---|---|---|---|---|---|---|---|---|---|---|
| | Q1 | O1 | P1 | K1 | N2 | M2 | S2 | K2 | RSS | M4 |
| FES12 | 0.216 | 0.309 | 0.355 | 0.471 | 0.342 | 0.658 | 0.407 | 0.223 | 1.120 | 0.115 |
| TPX08 | 0.153 | 0.310 | 0.181 | 0.442 | 0.201 | 0.523 | 0.338 | 0.151 | 0.893 | 0.069 |

| RMS Model Differences (cm) Against Shelf Water Tide Stations | | | | | | | | | | |
|---|---|---|---|---|---|---|---|---|---|---|
| European Shelf | | | | | | | | | | |
| FES12 | 0.88 | 0.82 | 0.71 | 1.19 | 1.39 | 3.71 | 1.94 | 0.63 | 4.82 | 2.22 |
| TPX08 | 0.88 | 0.72 | 0.46 | 1.21 | 1.58 | 3.85 | 1.70 | 0.74 | 4.87 | 0.35 |
| Elsewhere | | | | | | | | | | |
| FES12 | 0.80 | 1.00 | 0.89 | 1.51 | 1.58 | 3.33 | 2.30 | 1.02 | 4.96 | 0.98 |
| TPX08 | 0.82 | 1.00 | 0.82 | 1.47 | 2.00 | 3.50 | 1.93 | 1.12 | 5.07 | 0.88 |
| RMS Model Differences (cm) With 56 Coastal Tide Gauges | | | | | | | | | | |
| FES12 | 0.32 | 0.89 | 0.61 | 1.65 | 1.74 | 6.60 | 2.27 | 0.77 | 7.50 | 1.49 |
| TPX08 | 0.43 | 1.13 | 0.93 | 2.01 | 3.34 | 15.65 | 7.79 | 2.21 | 18.10 | 1.68 |

In the deep ocean and shelf regions, FES12 and TPXO8 have comparable RMS and RSS. In the coastal regions, TPXO8 is less accurate than FES12. We agree that the RMS statistic is probably not a good general descriptor, since one or two poor performing stations can dominate the results. But the results show that they have comparable accuracy even though TPXO8 has higher accuracy than FES12. Therefore, we use FES2014, which is the successor of FES12, as observations in this study.

- On p. 5 lines 130-135: Need to explain the advantages and disadvantages of DUD compared the other data assimilation algorithms.

Response:

DUD is one of the parameter estimation algorithms working in an iterative ensemble approach. Its advantages and disadvantages are:

1) Compared to the variational data assimilation algorithms, DUD is a method similar to Gaussian-Newton but derivative-free. The derivative-free approach can reduce the complexity of the estimation process.
2) DUD perturbs each parameter to generate the ensemble. While other ensemble algorithms usually estimate with an ensemble size smaller than the number of parameters and subsequently with a limited estimation accuracy due to the small ensemble size. However, DUD is not suitable for the system with a large number of parameters.

We add some description in the Section 2.2 "Parameter Estimation Scheme" of the Manuscript:

"Compared to the variational data assimilation algorithms, the derivative-free approach in the DUD can reduce the complexity of the estimation process. The size of ensembles in DUD is equal to the number of parameters, that ensures sufficient degree of freedom for parameter estimation, while other ensemble algorithms normally use an ensemble size smaller than the number of parameters and subsequently leads to a limited estimation accuracy. However, DUD is not suitable for the estimation with large number of parameters." (Highlighted in Page 5)

- Figure 1a: If possible, in Figure 1a, the authors need to put numbers used in y-axis of Figure 1b as area identification number. - Figure 1b: put titles of x-axis and y-axis.

Response: We have added the region identification numbers and polygons in Figure 1. The titles of x-axis and y-axis in Figure 1b are also added (Page 8). The updated figure is as follows:

[Figure]

Figure 1. Bottom friction energy dissipation in initial GTSMv4.1 (a) Global distribution [Unit:$W/m^2$]; (b) Area identification; (c) Area-integrated energy dissipation [Unit:TW].

- On p. 9 lines 216-219: The authors need to rewrite the sentences. Is there any reason to choose the specific year of 2014? Did you predict tides of 2014 along with tidal harmonic analyses?

Response: The selection of the year 2014 is based on the analysis of tide gauge data. The available tide gauges vary in different years. The reason for choosing 2014 is for it has the largest number of stations with available data. In this paper, GTSM is simulated over the whole year of 2014 for the tidal harmonic analysis.

We rewrite the paragraph in the Section 2.3.2 Observation Network, as follows:

"We select the year 2014 for the model analysis because the available tide gauges varying in different years and year 2014 has the largest number of stations. Tide analysis is performed in the tide gauge data from CMEMS and UHSLC dataset for the year 2014 with the TIDEGUI software, a matlab implementation of approach by Schureman (1958) and we visual inspect the tide and surge representations. After the analysis and quality control, we obtained 237 locations in the UHSLC dataset and 297 locations from the CMEMS dataset. In the deep ocean, we generate about 4000 time series from the FES2014 dataset to ensure enough observations for

estimating bathymetry in the year of 2014. These observations are evenly distributed and located in the deep ocean with a depth larger than 200m." (Highlighted in Page 10)

5. Section 3 (Estimation of Bottom Friction Coefficient)

- Figure 3a: What do the numbers (1, 2, and 3) in Figure 3a mean?

Response: The numbers (1,2 and 3) are just used to indicate the three subdomains we defined in the Hudson Bay/Labrador areas. It simplifies the reference in the discussion.

- On p. 12 lines 262-263: The authors need to put names including Foxe Basin, Hudson Strait and Ungave Bay in a location map.

Response: We have added the region names into the caption of this Figure (Figure 3 in page 14):

[Figure]

Figure 3: (a) Bottom friction energy dissipation per square meter of the Hudson Bay/Labrador in GTSMv4.1 [unit:W/m$^2$] and bottom friction coefficient subdomains (red boxes). Subdomain 1: Canadian Archipelago; Subdomain 2: the combination of Foxe Basin, Hudson Strait and Ungave Bay; Subdomain 3: Hudson Bay; (b) FES2014 observation distribution: Points in different subdomains have different colors.

- On p. 12 lines 264-269: The authors should rewrite these sentences to make them clear. What kind of "parameters" do you mean? What is "the form of tide components"? Does it mean "harmonic constants for tidal constituents"? How long do you use "model output of time series"?

Response: The parameters in this section are the bottom friction coefficient. "The form of tide components" means the harmonic constants for tidal constituents. The time series used for harmonic tide analysis is one year (the year of 2014) and for the estimation 1 month model simulation is used. We have reshaped the sentences in the paper as follows:

"The available observations are from the arctic stations but only include four major tidal components. In theory, harmonic tide analysis can be performed for the model output and it is possible to estimate parameters with the model output of harmonic constants for tidal constituents, but accurate tide analysis needs a time series of a year, which would increase the computation demand. For example, Wang et al., (2021b) reported that a full time series of 1 month is sufficient for an accurate parameter estimation. However, the yearly tide analysis would increase run times by a factor of 12. This is not feasible for us at the moment. Therefore, we choose to use the model output of time series covering 1 month in the estimation process and 1 year for harmonic tide analysis. The arctic stations can be used for the model validation." (Highlighted in Page 13)

- On p. 12 lines 284-285: The authors need to put names such as Scotland, the Faro Islands and Shetland in a location map. There were twice "The region of Scotland, the Faro Islands and Shetland have mountainous". Remove one.

Response: Sorry, it is a mistake. We have corrected it and put the names are in the figure (Figure 4 in page 15).

[Figure]

Figure 4. (a) Bottom friction energy dissipation per square meter across the European Shelf in GTSMv4.1 [unit:W/m²] and bottom friction coefficient subdomains (red boxes). Subdomain 1: The combination of the Scotland, Faro Islands and the Shetland; Subdomain 2: Irish Sea; Subdomain Subdomains 3 and 4: North Sea; Subdomain 5: English Channel; (b) CMEMS observation distribution: points in red are data used for calibration, points in green are used for validation and points in blue are not used.

6. Section 4 (Numerical Experiment and Results)

- On p. 14 lines 312-319: I think that these sentences were mentioned in previous sections.

Response: Thank you for your suggestion, we removed the repeated information and reshaped paragraph as follows:

"We selected a period of one month, September 2014 for the estimation runs. We found that simulation time length covering one month is sufficient for tide calibration when using high-frequency time series with 10 minutes sampling (Wang et al., 2021b)."

- On p. 14 line 315: Is there any reason to select "September" and "2014" for a period of one month?

Response: The selection of the year 2014 is based on our analysis of the largest number of available stations. And the month "September" is because the sea ice model is not included in the GTSMv4.1 and in September, there is no ice coverage so we can ignore the effect of sea ice.

We have added a description of this into the paper:

"In addition, sea ice in the Arctic Ocean is not modeled in the GTSM, but it has seasonal changes to the tides (Inger et al., 2021). Performing the experiment in the September can minimize the impact of sea ice to the model because of no ice coverage in the September." (Highlighted in Page 16)

- On p. 15 line 328: Are there any reason or reliable source to give the values of 5% and 20% uncertainty for bathymetry correction factor and bottom friction coefficient, respectively?

Response: Bathymetry is considered uncertain here because only a fraction of the ocean seabed has been surveyed, and the remaining errors are significant. Tozer et al. (2019) reported an estimate uncertainty of 150m for deep water and 180m between coastlines and the continental rise for the SRTM15+ dataset. Weatherall et al. (2015) showed in their Figure 6 the percentage of bathymetry changes between GEBCO_2014 and GEBCO_08 (GEBCO 2010 release) grids in the North Sea region with differences of over 5% or even 10% in many places. Therefore, the bathymetry uncertainty in this study is defined as 5%.

The Chezy coefficients of bottom friction are often empirically defined. A typical Chezy coefficient value is $62.5\ m^{1/2}s^{-1}$. The Chezy constants vary because of the ocean bed topography. For example, a higher value of coefficient is expected in the region with ocean mountain bathymetry. We use the value of 20% as the uncertainty of Chezy coefficient. The 20% changes of bottom friction coefficient is comparable to the 5% changes of bathymetry.

We have added the explanation into the Section 4.1.1 Experiment Design of the paper:

"Bathymetry uncertainty is defined as 5% from the knowledge that only a fraction of the ocean seabed has been surveyed, and the remaining errors are significant (Tozer et al., 2019). We empirically defined the uncertainty by investigating its varying range." (Highlighted in Page 16)

- On p. 18 line 357-359: There were twice "It is observed that in the Arctic Ocean, the initial RMSE with the value of 11.03cm is larger than other regions.". Remove one.

Response: We have corrected it.

**RC2: Referee William Pringle**

**General comments:**

This study uses a parameter estimation methodology implemented in an unstructured mesh global tide and surge model (GTSM v4.1) to estimate bathymetry and the bottom friction coefficient to reduce modeled tide errors at the coast. The parameter estimation methodology was developed by the authors in Wang et al. (2021, 2022), which focused on computational efficiency and memory efficiency of the parameter estimation algorithm by using model order reduction in space (Coarse Incremental Calibration) and in time (Proper Orthogonal Decomposition onto principal modes of variation). In those previous works the authors focused on perturbations to bathymetry to improve tide solutions. Therefore, the predominant novelty of this study is the simultaneous perturbation of the spatially varying bottom friction coefficient along with the bathymetry in a global model to assimilate tide observations and estimate these two parameters.

Although the tide errors of GTSM v4.1 are small and reasonable, I think the manuscript needs to do a better job of discussing why the errors in this model cannot be made as small as FES2014. Precisely what is the difference in data assimilation (DA) methodology that makes the TPXO/FES-type DA models able to give more accurate results overall than the parameter estimation technique used here? Also, what are the remaining major obstacles to further reducing tide error using the presented parameter estimation technique?

One of the reasons outside inaccurate bathymetry and unknown dissipation parameters for tide solution discrepancy could be errors associated with hydrodynamic simulation of the tide without concurrent simulation of meteorological-driven flow (surge). In shallow waters the estimation of bottom friction coefficient could be quite different in certain regions if surge is included due to nonlinear interaction. Furthermore, two recent related studies by the authors (Wang et al., 2021, 2022) also just investigate tide-only simulation, so to bolster this study the authors should consider adding in simulation(s) with meteorological forcing to show the sensitivity of tide solutions to concurrent surge simulation, especially since one of the main stated advantages of GTSM over FES/TPXO is the ability to simulate tide and surge together ("combined tide and surge model").

The other comment I have is on subdomain selection. In this study the two regions selected, Hudson Bay and European Shelf, are based on high tidal dissipation, which makes some sense. However, it is not clear how the subdomains within those regions are selected, although it appears to based on the authors' intuition (Line 284: "The region of Scotland, the Faro Islands and Shetland have mountainous ocean bathymetry, where expect to a higher bottom friction coefficient"). Have the authors investigated sensitivity to subdomain selection/size? Perhaps a spatial clustering type analysis or other could be used to more objectively find the suitable subdomains.

Response: Thank you for the valuable comments.

The possible reasons that our estimation is slightly worse than FES2014 are: The estimation in FES2014 gives an accurate estimate of the tide, but the underlying model is only used as a first guess or weak constraint. As a result of this choice, the tidal solution can be very accurate, but the result is a relatively static dataset. In contrast, the calibration of GTSM in this paper uses the model as a strong constraint. This results in a calibrated model that can used as a regular non-assimilative hydrodynamic model. For example we use the GTSM model for storm-surge forecasting and studying the impact of sea level rise; both are not possible with FES2014. Moreover, in our paper we aim to provide the solutions for accurate estimation when there is a lack of sufficient observations. Therefore, FES2014 uses more observations such as

satellite altimetry data, tide gauge data into the estimation. But GTSM use the FES2014 dataset as the observations in the deep ocean which sharply reduces the complexity of data preprocessing. Description has been added into the sections Introduction and Conclusion of the manuscript.

To better assess the estimation performance, we compared model performance and the FES2014 with the Bottom Pressure Recorder (BPR) gauge data in the deep ocean. FES2014 is slight better than GTSM and the main reason could be FES2014 uses BPR data in the assimilation process while BPR dataset is independent to our estimation procedure. In the shallow water we use part of the CMEMS dataset as observations and some FES2014 data in the region of scarce stations. FES2014 assimilated 600 tide gauges with a relatively homogeneous geographical distribution. We observed that FES2014 in the shallow water cannot perform as accurate as that in the deep ocean (Stammer et al., 2014). Therefore, it will also affect the estimation accuracy. In addition, FES2014 also corrected the internal tide coefficient and LSA (load and self-attraction) in the estimation process. When comparing with the estimation technique, FES2014 used the SpEnOI (Spectral Ensemble Optimal Interpolation) data assimilation algorithm and we use the computation and memory efficient DUD algorithm. Since estimation performance depends on many factors such as the number of observations, parameter size and so on. We don't think we can directly compare the assimilation techniques.

We add the major obstacles to further reduce tide errors in the conclusion of the paper:

"To further reduce tide errors using the presented parameter estimation technique, some major obstacles remain (1) When we consider to include satellite altimetry data especially in the shallow water to the estimation process, the accuracy of harmonic tidal analysis to the satellite altimetry has to be assessed, which would require complex preprocessing. (2) The influence of sea ice on the tide is currently not yet included into the model. However, the seasonal modulation from sea ice can affect the model performance (Kagan & Sofina, 2010; Müller et al., 2014), because sea ice exerts additional frictional stress on the surface. In our parameter estimation experiment, we observed that in the Canadian archipelago, higher bottom friction coefficients are estimated. This is probably caused by a lack of dissipation by sea ice. However, the estimated bottom friction coefficients do not result in a good agreement with the seasonal dynamics. A possible solution is to include the sea ice modeling in the GTSM, and the sea ice coefficient will also become an uncertain source to estimate. This will also require measurements that properly represent modulation of the tides over the seasons. Preliminary products of this type are starting to appear (Bij de Vaate et al., 2021). " (Highlighted in page 32)

Secondly, we add an experiment to compare the model derived surge before and after the estimation with the tide gauge data (UHSLC and CMEMS data). The surge simulation is performed for the year 2014 with the meteorology forcing (wind and air pressure conditions) from ERA5 reanalysis data set. The spatial-averaged monthly standard deviation of surge and total water-level simulation is shown in the following figure.

[Figure]

[Figure]

Figure (a) Spatial average STD between GTSMv4.1 and tide gauges for surge simulation in 2014; (b)Spatial average STD between GTSMv4.1 and tide gauges for total water level simulation in 2014 [Unit:cm]; CMEMS data includes 165 points (70 points are used in the calibration process and 95 points are for validation).

Please pay attention: We have to mention that in our author reply to reviewer 2 in the open discussion process, we provide a figure comparing with all the 297 observations from CMEMS dataset. But because a large amount of locations are not in the open ocean and we have removed these points from the CMEMS dataset in the paper. Finally, in this paper only 165 locations are used for calibration and validation. Therefore, for consistency, I redraw this figure by comparing the model performance with only these 165 points.

We add the following paragraph to the Section Numerical Experiment and results of the paper:

 "The standard deviations before and after the estimation show minor difference in Figure 11a. It is consistent with findings in our previous research to estimate bathymetry for GTSM (Wang et al., 2021). The error are generally larger in the areas with stronger tide in the shallow waters. This makes the absolute value of the STD very dependent on the tide gauges that are used. In the UHSLC dataset, the locations are spread over the planet. The CMEMS dataset focuses on the European Shelf with stronger winds in winter.

In general, surge comparison shows that surge is not sensitive to the bathymetry and bottom friction but strongly affected by the wind and air pressure conditions. This conclusion is also supported by Chu et al., (2019) to access the sensitivity of surge in the east China Sea (their Figure 13).  In our study, even though surge simulation keeps the same accuracy after the

estimation, the water level forecast accuracy is improved because of the improvement of tide representations, which is significantly demonstrated in the Figure 11b. Therefore, the bottom friction and bathymetry estimation improves the model derived water level forecast ability in the coastal areas." (Highlighted in page 25)

Moreover, thanks for your suggestion. It is a good idea to use a spatial clustering type analysis to find the suitable subdomains for parameters in the future. In this study, our selection of bottom friction subdomain is based on the following three rules: (1) subdomain is in the region with large tide energy dissipation; (2) subdomain is selected in which the observations are well-distributed. (3) Considering the sea bed topography, subdomain is set to divide the region that would have different values of coefficient.

Point-by-point comments:

Line 52: "We found only one application [of data-assimilation to estimate parameters] at a global scale (Lyard et al., 2021)…". Although it is a very recent study available as a pre-print, Blakely et al. (2022) also tries to "optimize" parameters for internal tide and bottom friction in a global tide model using the TPXO tide solutions, which I think would be worth referencing and comparing to in this manuscript.

Response: We have included a reference to Blakely's paper into the Introduction of the Manuscript as follows:

"Blakely et al. (2022) adjusts the bottom friction and internal tides friction in 41 subdomains to better represent the tide in the ADCIRC model, allowing the bottom coefficient to vary with the subdomain gives a significant improvement on model performance." (Highlighted in page 2)

Line 59: "The sensitivity to bottom friction is very small in deep water, but is often the most sensitive parameter in shallow water". Can the authors find some reference(s) for this? For one, I suggest Zaron (2017) here who presents a friction number that denotes the relative importance of the friction parameter in the momentum balance, and I think Zaron's paper will also provide material that can be used to improve the ideas presented in this part of the introduction.

Response: Thank you for the suggestion. We have added a reference to the paper of Zaron into the Introduction of the Manuscript as follows:

"Zaron (2017) denoted the relative importance of the friction parameter in the momentum balance in the Sea of Okhotsk based on the sensitivity test results." (Highlighted in page 3)

Section 2.1: There are numbers quoted for the tidal energy dissipation, 3.7 TW; 2.39 TW for bottom friction and 1.12 TW for internal tides. Do these numbers always stay constant no matter the bathymetry and bottom friction parameters being estimated? I also suggest to put these numbers in context with other tidal dissipation values from the literature as well to give an idea to the reader of the typical ranges and inter-model variability.

Response: The values of tidal energy dissipation before and after the estimation are consistent. After the estimation, the tidal energy dissipation is 2.44TW for the bottom friction and 1.33TW for the internal tides. The total tidal energy dissipation is 3.77TW. It implies that the estimated bottom friction coefficients are reasonable. Moreover, this value also matches the findings of other researchers with a global dissipation of around 3.7TW either from the model simulation or measurement analysis (Egbert and Ray, 2001; Green and Nycander, 2013; Munk and Wunsch, 1998).

We added the description into the section Parameter to Estimate of the paper as follows:

"The value of tide energy dissipation matches the findings of other researchers with a global dissipation of around 3.7TW either from the model simulation or measurement analysis (Egbert and Ray, 2001; Nycander, 2005; Munk and Wunsch, 1998)." (Highlighted in page 8)

Line 111: "[The Chezy formulation] is important for hydrodynamic conditions". What does this mean?

Response: We rewrite the sentence as: "Bottom friction term is important to determining hydrodynamic conditions and sediment transport process." (Highlighted in page 5)

Lines 114-116. These statements require more detail. Exactly how is the internal tide friction term corrected for layer thickness in the salinity/temperature dataset (what does this mean?). How was the retweaking of the bottom friction and internal tide coefficients done and how does this compare to this study which is trying to find improved bottom friction coefficients?

Response:  We rewrite the sentence in the section Global Tide and Surge  Model of the paper as follows:

"GTSMv4.1 contains an updated internal tide friction term that is related to the buoyancy frequency of the stratified ocean. In the previous version of GTSM, the layer thickness variability was not taken into account properly and this was fixed in the dataset for GTSMv4.1. The correction coefficient for this improved dataset is derived again and the spatial uniform bottom friction was set to a value found more often in literature." (Highlighted in page 5)

Lines 118-119: States the RMSE is without the bias difference. Does just mean the RMSE used here is the standard deviation of the error? I notice Figure 9 panels have the title of "Standard Derivation …" which maybe should read standard deviation. Please clarify.

Response: We remove the bias difference from model and observations. Because the simulation is only related to tide and the mean difference could be from the different reference plane of observation. We also corrected the title of Figure 9 to be "standard deviation".

Lines 128-129: "However, the spectral tide model cannot describe the interaction between different tide components in shallow waters." What is meant by "describe" here? In Le Provost & Lyard (1997), which is the underpinning of the FES model, the methodology considering tide component interaction through linearization of the bottom friction term is presented. So while it's true that the tide component interaction in a spectral model cannot be computed "exactly" like in a time-stepping shallow water model, some interaction through the bottom friction term can be accounted for.

Response:  we have rephrased the sentence in the paper as follows:

"However, the spectral tide model cannot exactly compute the tide components interaction, even though some methodologies such as representing the interaction through linearization of bottom friction term are presented (Le Provost & Lyard, 1997)." (Highlighted in page 5)

Section 4.1.2: Parameter estimation results: Only relative changes to the parameters are shown but I think it would be interesting information for readers to know the initial and final values of the bottom friction coefficients (which may be compared to bottom friction values obtained in Blakely et al., 2022).

Response: The initial value of the bottom friction coefficient is 62.5. After the estimation, the bottom friction values are shown in the following figure.

[Figure]

Figure: Bottom friction coefficients after the parameter estimation of GTSM.

We didn't compare the difference of bottom friction values between our study and that in Blakely et al., 2022 because we use the Chezy formula while Blakely use the Manning formula. The coefficient C in the Manning formula is related to the depth H as $C = {H^{1/6}}/{n}$.

Lines 491-492: "Tide representation in shallow waters benefits from the optimization of bottom friction coefficient, contributing to a more accurate water level forecast when including wind and air pressure conditions for surge simulation". This is more than likely correct but is not a conclusion that can be straightforwardly made from the study. If, as I mention in the general comments, this study considers the sensitivity of the parameter calibration to tides with concurrent simulation of surge, it should help to provide stronger evidence for this statement.

Response: We compared the model simulated surge before and after the estimation, the results are shown in the reply to the general comment.

Technical corrections:

Line 126: What is SLA?

Response: It is the Self-attraction and loading. In Lyard et al. (2021), they call it loading and gravitational self-attraction (LSA). I have rephrased the sentence in the Global Tide and Surge Model of the paper as follows:

"FES2014 uses the Spectral Ensemble Optimal Interpolation (SpEnOI) algorithm to estimate the bottom friction coefficient, the internal tide drag coefficient, the bathymetry and the LSA (loading and gravitational self-attraction). It leads to an accurate data collection of 34 tidal components." (Highlighted in page 5)

Table 1/Line 127: TPOX09 to TPXO9.

Response: corrected.

Lines 355-369: In these two paragraphs a confusing terminology of the RMSE being reduced to X% is used. I think it's easier to understand how much the RMSE was reduced BY.

Response: We have rephrased these sentences using "reduced by".

References:

Blakely, C. P., Ling, G., Pringle, W. J., Contreras, M. T., Wirasaet, D., Westerink, J. J., Moghimi, S., Seroka, G., Shi, L., Meyers, E., Owensby, M., & Massey, C. (2022). Dissipation Processes in an Unstructured Mesh Global Tidal Model. Under review at Journal of Geophysical Research: Oceans. https://doi.org/10.1002/essoar.10509993.1

Le Provost, C., & Lyard, F. (1997). Energetics of the M2 barotropic ocean tides: an estimate of bottom friction dissipation from a hydrodynamic model. Progress in Oceanography, 40(1), 37–52. https://doi.org/10.1016/S0079-6611(97)00022-0

Wang, X., Verlaan, M., Irazoqui Apecechea, M., & Lin, H. X. (2021). Computation-Efficient Parameter Estimation for a High-Resolution Global Tide and Surge Model. Journal of Geophysical Research: Oceans, 126, e2020JC016917. https://doi.org/10.1029/2020JC016917

Wang, X., Verlaan, M., Irazoqui Apecechea, M., & Lin, H. X. (2022). Parameter Estimation for a Global Tide and Surge Model with a Memory-Efficient Order Reduction Approach. Under review at Ocean Modelling.

Zaron, E. D. (2017). Topographic and frictional controls on tides in the Sea of Okhotsk. Ocean Modelling, 117, 1–11. https://doi.org/10.1016/j.ocemod.2017.06.011

Chu, D. , Zhang, J. , Wu, Y. , X Jiao, & Qian, S. . (2019). Sensitivities of modelling storm surge to bottom friction, wind drag coefficient, and meteorological product in the east china sea. Estuarine, Coastal and Shelf Science, 231, 106460-.

---

## Author Response (AR2)

**Many thank you for your time, comments and minor linguistic rectifications to improve the quality of this paper.**

The latest version of the paper has been reviewed, and we are in agreement that it can now proceed. I agree with the reviewer's correction about a small edit to line 321. Please can you do that, and then the paper can go forward for typesetting.

Response: Thank you for the valuable comment. We have changed the sentence to ""surge is not sensitive to small random perturbations of the bathymetry".

Also note, Eqn 3 is incorrect (unbalanced brackets).

Response: corrected.

There are a couple of minor grammar errors, eg arctic ->Arctic (several instances), line 519 cut "consider to".

Response: We have checked and corrected these grammar errors.

Please fill in the missing dois in the references.

Response: All the dois are added in the references.

Non-public comments to the Author:

I'm not sure if you've received the report from the reviewer, just in case here it is!

>>I thank the authors for responding to my comments.

In particular, the authors clearly show that the surge is not sensitive to the parameter estimation technique but tides and total water levels are, thereby justifying their approach to focus on parameter estimation for the tides only. I would just add that I think this is because the coastal tide gauges used here are ones located along open coastlines. For higher-resolution in more complex estuarine and river regions I would suspect that we would start to see more impact on the surge and the nonlinear interaction between surge and tide.

Response: Thank you for the valuable comment. We have added the sentence into the paper:

"Moreover, we are expected to research on the impact on the surge and the nonlinear interaction between surge and tide for higher-resolution in more complex estuarine and river regions in the future."

>>I have one suggested technical edit:

Line 321: "surge is not sensitive to the bathymetry", change to "surge is not sensitive to small random perturbations of the bathymetry".

The water depth is very important for determining the size of the surge so I think the phrasing that surge is not sensitive to bathymetry in an absolute sense is wrong.

Response: We changed this sentence in the paper.